# Amphibian gut microbiota shifts differentially in community structure but converges on habitat-specific predicted functions

Molly C. Bletz[1], Daniel J. Goedbloed[1], Eugenia Sanchez[1], Timm Reinhardt[2], Christoph C. Tebbe[3], Sabin Bhuju[4], Robert Geffers[4], Michael Jarek[4], Miguel Vences[1] & Sebastian Steinfartz[1]

Complex microbial communities inhabit vertebrate digestive systems but thorough understanding of the ecological dynamics and functions of host-associated microbiota within natural habitats is limited. We investigate the role of environmental conditions in shaping gut and skin microbiota under natural conditions by performing a field survey and reciprocal transfer experiments with salamander larvae inhabiting two distinct habitats (ponds and streams). We show that gut and skin microbiota are habitat-specific, demonstrating environmental factors mediate community structure. Reciprocal transfer reveals that gut microbiota, but not skin microbiota, responds differentially to environmental change. Stream-to-pond larvae shift their gut microbiota to that of pond-to-pond larvae, whereas pond-to-stream larvae change to a community structure distinct from both habitat controls. Predicted functions, however, match that of larvae from the destination habitats in both cases. Thus, microbial function can be matched without taxonomic coherence and gut microbiota appears to exhibit metagenomic plasticity.

[1] Zoological Institute, Technische Universität Braunschweig, Mendelssohnstr. 4, Braunschweig 38106, Germany. [2] Institute for Zoology and Cell Biology, Heinrich-Heine-Universität Düsseldorf, Universitätsstraße 1, Düsseldorf 40225, Germany. [3] Thünen Institute of Biodiversity, Bundesallee 50, Braunschweig 38116, Germany. [4] Helmholtz Centre for Infection Research, Department of Genome Analytics, Braunschweig 38124, Germany. Correspondence and requests for materials should be addressed to M.C.B. (email: molly.bletz@gmail.com) or to S.S. (email: s.steinfartz@tu-bs.de).

The vertebrate digestive system is inhabited by diverse microbial communities[1,2], which play a fundamental role in the well-being of their host[3,4]. A host's gut microbiota mediates many processes such as digestion and energy acquisition[5,6], vitamin synthesis[7], immunomodulation[8] and pathogen defence[9–11], and may even be an important factor in processes of ecological adaptation[12].

Multiple factors and processes shape the composition of these symbiotic host-associated communities[13]. Host factors such as host genotype, stomach pH, mucins and antimicrobial peptides can impose selective filters thus shaping community composition[13–18]. External factors such as diet and surrounding environment are also known to substantially influence microbial community composition[19–32]. Diet itself may strongly select the microbial community for its ability to degrade specific molecules[27,32], and diet-associated microbes may also represent a source of potential colonists of the gut[28]. Abiotic environmental conditions, such as temperature, can also influence gut microbial community structure, especially in ectothermic organisms[29–31].

From an ecological perspective, the gut can be seen as a unique microbial habitat in which gut microbial diversity and structure can be explained by principles of classical island biogeography theory (that is, invasion, colonisation, immigration and extinction), and community ecology theory (that is, deterministic, neutral and historic processes of community assembly)[13,33,34]. Within this framework, entry into a new habitat by a host not only includes abiotic changes but also may involve alterations of a host's diet, and therefore, a host's gut microbial community can be predicted to shift in response to changing ecological conditions. Detecting such complex shifts in structural diversity of microbial communities independent of cultivation is possible via molecular tools that PCR-amplify and sequence marker genes, such as the bacterial 16S rRNA gene, through next-generation sequencing approaches[35]. These approaches can determine whether and how communities structurally respond to environmental change. In addition to analysis of the structural response, it is perhaps even more important to study whether such structural changes are coupled with functional adjustments. Functional diversity cannot be easily studied by marker gene amplification, and therefore shotgun metagenomics can be used to assess functional diversity. However, there are also bioinformatics approaches, such as Phylogenetic Investigations of Communities by Reconstruction of Unobserved States (PICRUSt) that allow the metagenome (for example, functional gene family abundances) of a community of organisms to be predicted from a phylogenetic marker gene, such as the 16S rRNA gene. Although such tools have limitations in comparison with shotgun metagenomic sequencing approaches, PICRUSt has proven to be a powerful tool for describing and comparing functional attributes of microbial communities[36–38].

Although knowledge of the impact of host factors on gut microbiota is increasing, limited understanding exists of the external environment's influence on these communities and of how these communities change under natural conditions. Experimental investigation of gut microbial community dynamics has mainly been completed in laboratory settings with model organisms. Currently, we are lacking an in-depth understanding of ecological dynamics of host-associated microbiota in natural habitats from a diverse host range.

Fire salamanders (*Salamandra salamandra*) in Western Germany provide a well-studied natural amphibian system, where females preferentially deposit fully developed larvae in two different habitat types (that is, small first order streams versus stagnant ephemeral ponds). This differential reproductive behaviour has been causing genetic adaptive divergence in certain populations[39–42]. The Streams and ponds in which the larvae

develop differ in many abiotic and biotic conditions[43]. While streams represent a stable environment with low temperatures and constant water supply throughout the year, ponds are less predictable with high variation in temperature, and larvae face a high risk of rapid desiccation[43]. These habitat types also differ in the composition and quantity of potential and consumed food resources, and restriction of food availability with starvation being typical for ponds and unknown from streams[39,44].

We used this salamander system to: (1) test the hypothesis that different microbial communities identified by 16S rRNA gene amplicon sequencing will characterize individuals from habitats with distinct environmental conditions (that is, ponds and streams) via a field survey and (2) test if and to what extent environmental change will cause a shift in gut microbial composition and/or community structure, and how this relates to predicted functional potential of these communities (determined by PICRUSt) utilising a reciprocal transfer experiment. The reciprocal transfer of individuals between different habitat types is a classical and powerful approach used in ecology and evolutionary biology to determine how and by which underlying mechanisms individuals adjust to and perform in specific environments. We used sampling of the skin microbial communities within the reciprocal transfer experiment as a reference for comparison. Amphibian skin microbiota is well-studied and known for its defensive function against pathogens[45,46]. Because these skin communities are considered to be influenced more directly by external abiotic parameters owing to direct exposure to the environment, they can act as a control when looking at the gut microbial communities which have additional nutritional (that is, diet) influences.

Our study offers important new insights into gut bacterial community dynamics of amphibians, an understudied host group, under natural conditions. We find that gut bacterial community structure of salamander larvae is habitat-specific, demonstrating that abiotic and biotic environmental factors mediate community structure. In addition, although skin communities shift in response to environmental change to simply match the community structure of the destination habitat controls, gut communities reach habitat-specific functionality by shifting in two distinct ways (matching and non-matching taxonomic shifts) depending on the directionality of transfer.

## Results

**Gut microbiota of salamander larvae differ between habitats.** Gut bacterial communities were investigated with 16S rRNA gene amplicon sequencing. Alpha diversity differed between pond and stream larva gut bacterial communities, with stream larvae harbouring greater Chao1 diversity values and greater phylogenetic diversity, PD ($df = 1$, Chao1: KW $\chi^2 = 12.398$ $P < 0.0004$, PD: KW $\chi^2 = 7.4624$, $P = 0.006$, Fig. 1a). Species evenness in these gut bacterial communities was similar between pond and stream larvae (Simpson's evenness: KW $\chi^2 = 0.28061$, $P = 0.596$, Fig. 1a). Approximately 30% of the OTUs in the larval gut community and only 11% of the OTUs in the Core50 communities (see Methods) were *de novo* OTUs (that is, OTUS clustered by a *de novo* algorithm in the QIIME open-reference clustering pipeline).

Beta diversity of gut bacterial communities differed between pond and stream larvae using the Bray-Curtis (PERMANOVA): Pseudo-$F_{(1,75)} = 7.24$, $P = 0.001$), weighted Unifrac (PERMANOVA: Pseudo-$F_{(1,75)} = 13.911$, $P = 0.001$, Fig. 1b), and unweighted Unifrac (PERMANOVA: Pseudo-$F_{(1,75)} = 6.022$, $P = 0.001$) metrics. Furthermore, the composition of these gut bacterial communities differed from that of the surrounding environment (Supplementary Fig. 1). Gut bacterial communities were dominated by major bacterial groups that differed in relative

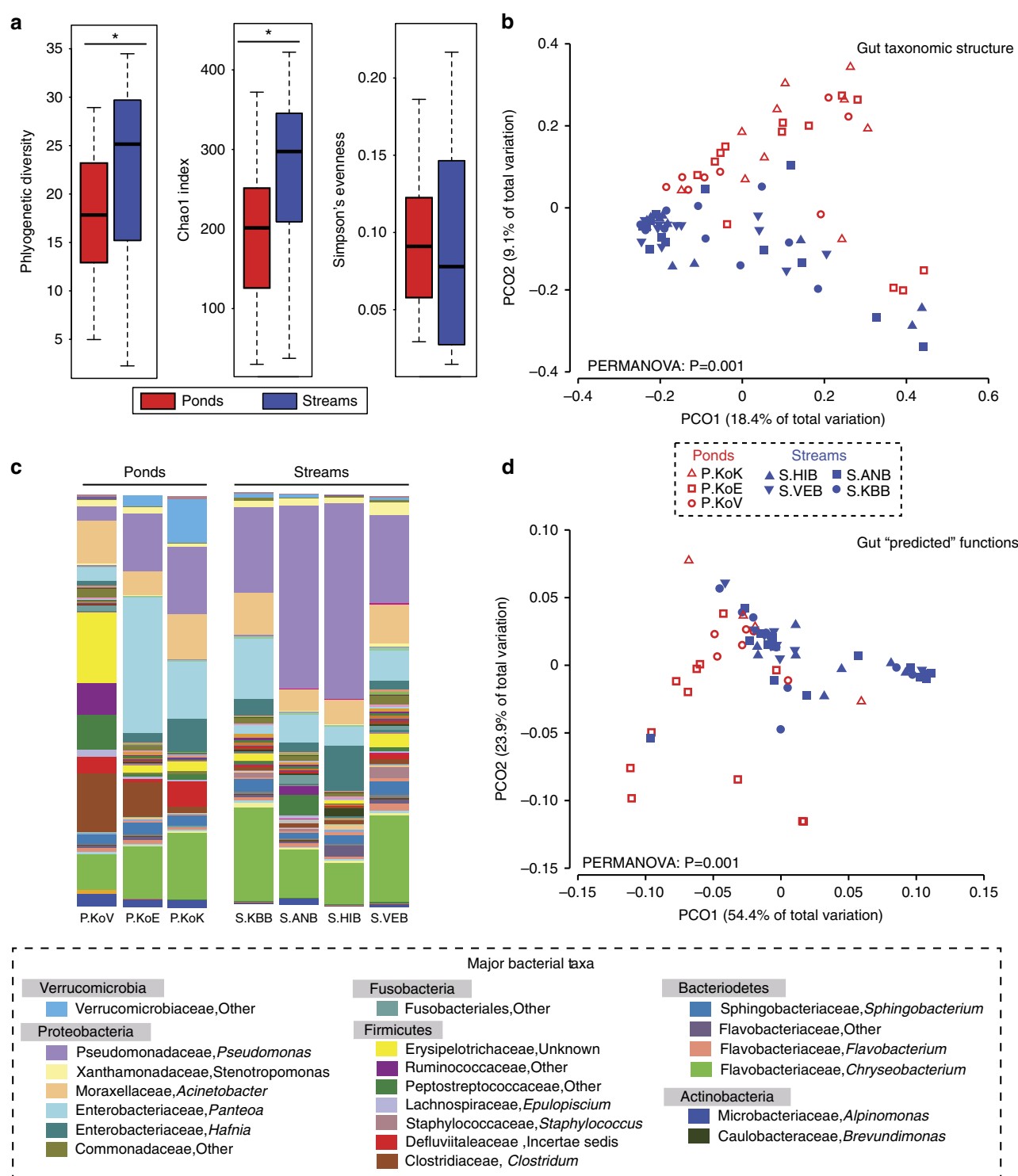

**Figure 1 | Pond and stream larvae exhibit distinct gut microbiota and predicted functions.** (**a**) Alpha diversity between pond and stream larvae for Phylogenetic Diversity and Chao1 and Simpson's Evenness Index. Asterisks denote statistically significant differences as determined by Kruskal–Wallis tests. (**b**) Principal Coordinates Analysis of unweighted Unifrac distance matrix showing separation of pond and stream larva gut communities. Main-effect PERMANOVA results are displayed. Permutational test of dispersions (PERDISP) showed no differences in dispersion between pond and stream communities (F $_{(1,75)}$ = 0.3161, $P$ = 0.625). (**c**) Mean relative abundance profiles of Core50 gut bacterial taxa (#OTUs = 82) at the genus level from four stream sites and three pond sites. Dominant taxa are identified in the legend. (**d**) PCo analysis of predicted functional profiles of gut bacterial communities from the field survey. Sample sizes were: $n$ = 46 (streams); $n$ = 31 (ponds). Pond larvae are shown in red and stream larvae are shown in blue in panels (**a**, **b** and **d**).

abundance between pond and stream larvae. The most dominant taxa were Proteobacteria (Alpha-pond: 4.9%, stream: 7.0%; Beta-pond: 2.3%, stream: 3.0%; Gamma-pond: 23.1%, stream: 50.9%),

Firmicutes (pond: 41.3%, stream: 10.7%), Bacteroidetes (pond: 14.1%, stream: 20.1%), Actinobacteria (pond: 3.9%, stream: 6.6%) and Verrucomicrobia (pond: 6.7%, stream: 1.4%). These groups

together made up on average 97.7% of the community. The observed differences in community structure were driven by differences in the relative abundance of multiple bacterial taxa (Fig. 1c). Overall, 26 differentially abundant taxa, ranging from the phylum to OTU-level, were detected with linear discriminant analysis (LDA) effect size (LEfSe) (LDA score >2). Eight of these taxa had greater relative abundances in pond larvae and 18 taxa had greater relative abundances in stream larvae (Supplementary Fig. 2). At the phylum level, Alpha- and Gammaproteobacteria were more abundant in stream larvae, whereas Firmicutes were more abundant in pond larvae (LEfSe LDA >2, Supplementary Fig. 2). At the OTU-level, four OTUs, including a *Pseudomonas* sp. (Gammaproteobacteria*), Methylobacterium* sp. (*Alpha*proteobacteria*), Agrobacterium tumefaciens* (Alphaproteobacteria) and *Veillonella* sp. (Firmicutes) were more abundant in stream larvae and two OTUs, including a Peptostreptococcaceae sp. (Firmicutes) and an *Alpinimonas* sp. (Actinobacteria) were more abundant in pond larvae. Relative abundances of these OTUs are shown in Supplementary Fig. 2. The Core50 communities of stream larvae and pond larvae were comprised of 72 and 30 OTUs, respectively, and 20 of these OTUs overlapped between these groups. These 20 OTUs included members of the Flavobacteriales (7), Pseudomonadales (5), Sphingobacteriales (3), Enterobacteriales (2), Verrucobacteriales (1) and Legionellales (1).

PICRUSt was used to predict functions of gut bacterial communities. Analysis of these predicted function revealed significant differences between the functional profiles of pond and stream larvae (PERMANOVA: Pseudo-$F_{(1,63)}$ = 11.69, $P$ = 0.001, Fig. 2d). In addition, 79 metabolism-associated functional features of the gut bacterial communities were differentially abundant between pond and stream larvae (LEfSe LDA >2.0). Thirty-six features were more abundant in pond larva bacterial communities and 43 were more abundant in stream larva bacterial communities (Supplementary Table 1). For example, pond larvae had multiple differential features associated with carbohydrate metabolism, including galactose metabolism, pentose and glucoronate interconversions, starch and sucrose metabolism, and fructose and mannose metabolism. Bacterial communities of stream larvae had multiple functional features associated with lipid metabolism including, lipid metabolism, fatty-acid metabolism and glyoxylate and dicarboxylate metabolism. Full lists of detected differential features are given in Supplementary Table 1.

**Larval diet differs between habitats**. The stomachs of pond individuals contained mainly microcrustaceans (ostracods, copepods, cladocerans), whereas those of stream individuals contained larger crustaceans (amphipods) and insects (coleopterans and dipteran larvae) in addition to microcrustaceans (ostracods and copepods) (Fig. 2). Stream larvae also ingested more detritus and unidentified particles.

**Taxonomic shifts of habitat-switched larva gut microbiota**. Gut bacterial communities of larvae that were transferred between habitats in the fully factorial reciprocal transfer experiment (Fig. 3) showed distinct responses. The gut community structure and composition of stream larvae transferred into ponds (SP) shifted to match that of pond-to-pond (PP) control individuals (pair-wise PERMANOVA: SP-SS: Average distance (Average Distance (AvgDist) = 0.613, t = 2.5303, $P$ = 0.001; SP-PP: AvgDist = 0.416, t = 0.80152, $P$ = 0.817, Fig. 4a,c). In contrast, the communities of pond larvae transferred into streams (PS) were distinct from both PP and stream-to-stream (SS) control individuals (pair-wise PERMANOVA: PS-PP: AvgDist = 0.527, t = 1.8343, $P$ = 0.005; PS-SS: AvgDist = 0.636, t = 2.2454, $P$ = 0.001, Fig. 4b,c). This divergent pattern (taxonomic convergence in the SP to PP and lack of matching in the PS to SS) occurred across all taxonomic levels (operational taxonomic unit (OTU) to phylum level) (Supplementary Fig. 3). Importantly, experimental enclosures did not affect gut bacterial community structure compared with free-swimming larvae (pair-wise PERMANOVA: S(free)-SS: t = 1.1033, $P$ = 0.225; P(free)-PP: t = 0.60039, $P$ = 0.996, Supplementary Fig. 4). Chao1 diversity slightly differed between experimental stream larvae and free-swimming stream larvae ($df$ = 1, Kruskal–Wallis (KW) chi-squared = 3.9455 $P$ = 0.047); however, this difference was not mirrored in phylogenetic diversity of stream larvae ($df$ = 1, KW $\chi^2$ = 1.1626 $P$ = 0.2809). Experimental pond larvae and free-swimming pond larvae did not differ in Chao1 diversity

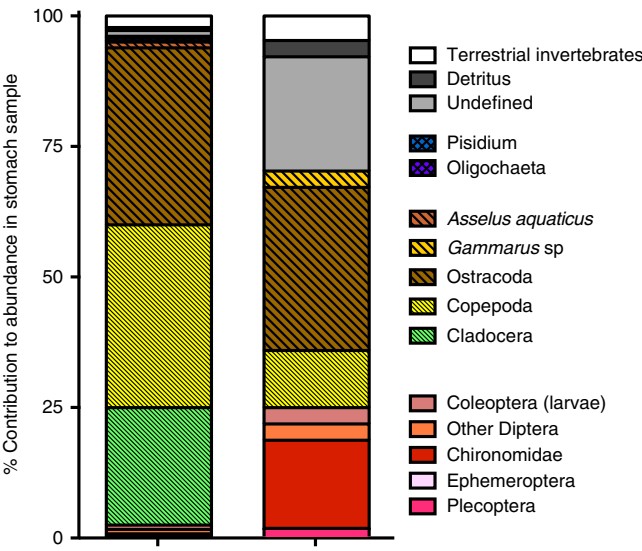

**Figure 2 | Food consumption differs between larvae from different habitats.** Proportional abundance of items in the stomach contents of pond and stream larvae sampled in the field survey. Sample sizes were: $n$ = 46 (streams); $n$ = 31 (ponds).

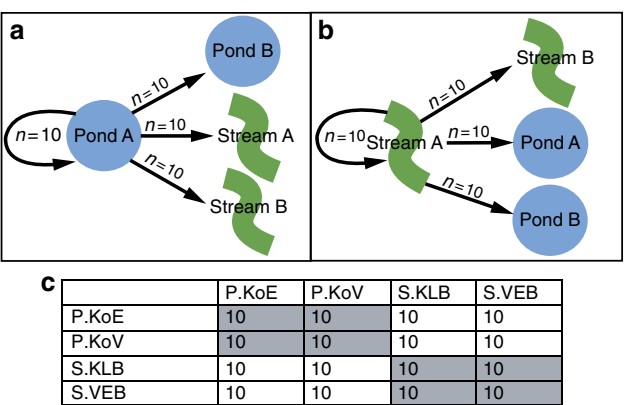

**Figure 3 | Experimental design of the reciprocal transfer experiment.** (**a**) Representation of transfer design for salamander larvae native to a pond site. (**b**) Representation of transfer design for larvae native to a stream site. (**c**) Experimental design cross-table with sample sizes; control groups are highlighted in greyed cells. P denotes pond sites and S denotes stream sites. Pond sites include, KoV and KoE and stream sites include, Vennerbach, VEB; Klufterbach, KBB.

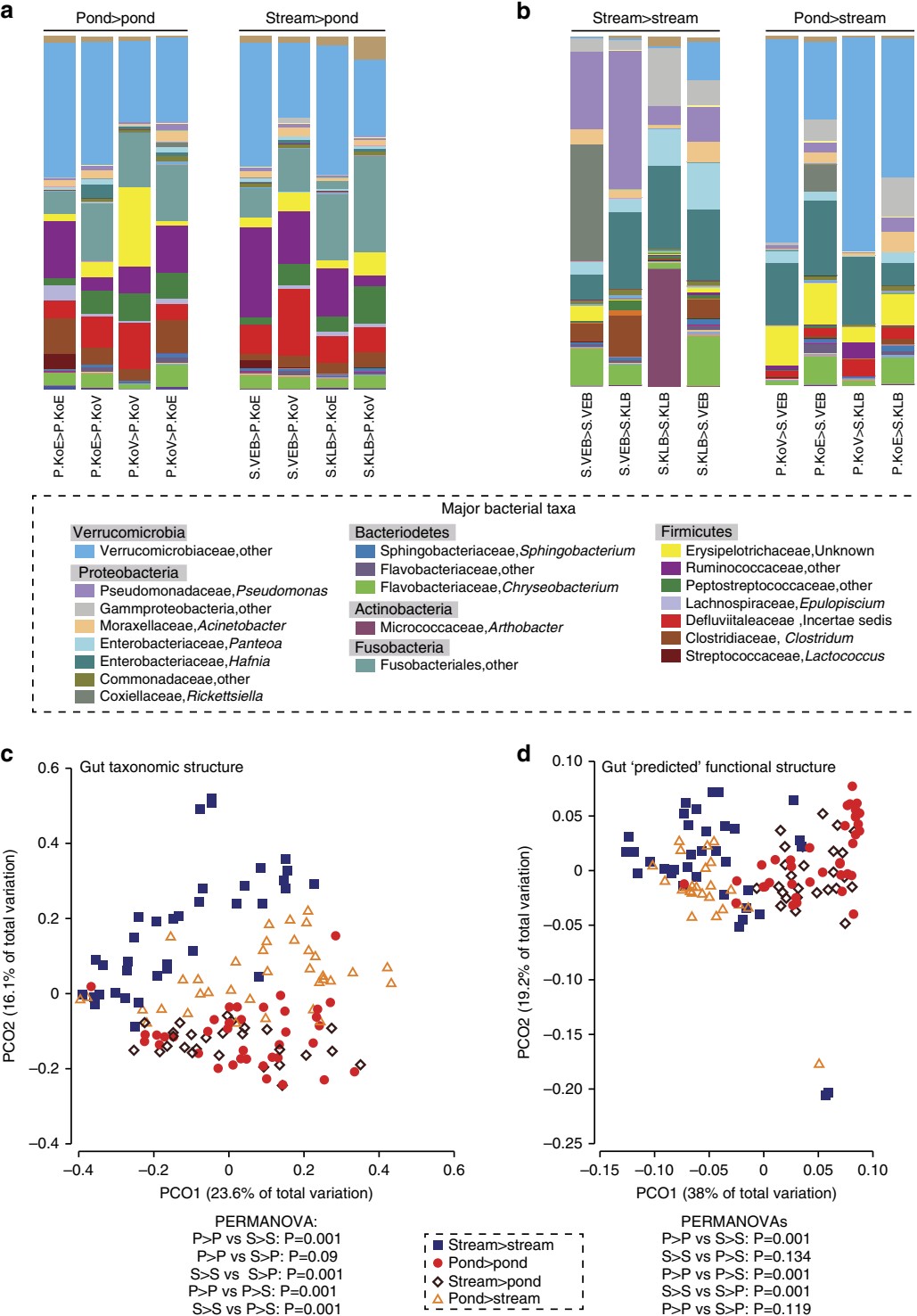

**Figure 4 | Gut bacterial community structure and predicted functions in response to habitat transfer. a b** show mean relative abundance profiles of Core50 OTUs in the gut bacterial communities of larvae at the genus level. **a** shows larvae derived from streams and transferred to ponds in comparison with PP controls. **b** shows larvae derived from ponds and transferred to streams in comparison with SS controls. **c** shows Principle Coordinate Analysis of unweighted Unifrac distance matrices of the Core50 communities of larvae in the transfer experiment. **d** shows PCo analyses of predicted functional profiles of gut bacterial communities from larvae in the transfer experiment. Functional PCoA visualisations are based on Bray-Curtis distance matrices. Sample Sizes were: $n = 36$ (SS, blue squares), $n = 38$ (PP; red circles), $n = 27$ (SP; brown open squares), $n = 36$ (PS; yellow open triangles). Pair-wise PERMANOVA results are displayed below each PCoA. PERMDISP showed that P > S differed in dispersion from S > S ($t = 5.0108 \, P = 0.001$), but S > P did not differ from P > P ($t = 0.80695 \, P = 0.431$). Note that larvae transferred from PS become distinct from PP individuals and SS individuals, (**b,c**) whereas larval transferred from SP fully diverge from the bacterial communities of SS individuals and become identical (that is, not significantly different from) to PP larvae (**a,c**). For the predicted functions, however, both transfer groups switched to make that of the destination habitat controls (**d**).

($df = 1$, KW $\chi^2 = 1.5474$ $P = 0.2135$) or phylogenetic diversity ($df = 1$, KW $\chi^2 = 0.18625$ $P = 0.6661$, Supplementary Fig. 4).

Given the hypothesis that gut bacterial communities of habitat-switched individuals may experience a shift in OTUs, LEfSe analyses were performed between each group of habitat-switched individuals and their respective habitat-control group (that is, PS was tested against PP and stream-to-pond (SP) was tested against SS). Accordingly, 32 differentially abundant taxa were identified between the SS (8 taxa) and SP groups (24 taxa) (Fig. 5, Supplementary Table 2). Members of the Enterobacteriales order and a *Pseudomonas* sp. were the main taxa that were more abundant in SS larvae, whereas bacterial taxa from the Fusobacteriales, Clostridiales, Erysipelotrichiales and Verrucomicrobiales orders were more abundant in SP larvae (Supplementary Fig. 5). Twenty-five differentially abundant taxa were found between the PS and PP groups (Supplementary Table 3). Bacterial taxa from the Enterobacteriales were more abundant in PS larvae, and taxa from the Fusobacteriales, Clostridiales and Micrococcales orders were more abundant in PP larvae (Supplementary Fig. 5).

**Convergence of predicted functions to destination habitats**. The predicted functions of gut bacterial communities of habitat-switched individuals differed from their origin habitat controls (pair-wise PERMANOVA: PS-PP $t = 5.163$, $P = 0.001$; SP-SS $t = 3.839$, $P = 0.001$), but did not differ from their respective destination habitat controls (PERMANOVA: PS-SS $t = 1.254$, $P = 0.134$; SP-PP $t = 1.240$, $P = 0.119$). The habitat controls also differed from each other (PERMANOVA: SS-PP $t = 4.432$, $P = 0.001$) (Fig. 4d). Furthermore, LEfSe analyses revealed a strong functional shift in the gut microbial communities of habitat-switched larvae. The SP group had only 15 differential features when compared with PP controls, whereas there were 145 differential features when compared with the SS group (Fig. 5). The PS group when compared with the SS control group had only 24 differential functional features, whereas 154 of such features were detected between the PS and PP groups (Fig. 5).

**Taxonomic shifts of habitat-switched larva skin microbiota**. In general, the cutaneous communities found on salamander larvae were characterized by bacterial groups that are typical for amphibian skin, including Proteobacteria, Bacteriodetes and Actinobacteria (a detailed characterisation of the skin microbiota

of fire salamander larvae is provided elsewhere[47]). The skin community structure and composition of habitat-switched larvae in both cases shifted to match that of the destination habitat-control individuals (unweighted Unifrac, pair-wise PERMANOVA: SP-SS: $t = 1.6927$, $P = 0.001$; SP-PP: $t = 1.0604$, $P = 0.238$, PS-PP: $t = 1.8698$, $P = 0.001$; PS-SS: $t = 0.9791$, $P = 0.467$, Fig. 6). PP larvae also differed from SS controls (pair-wise PERMANOVA: PP-SS: $t = 1.6927$, $P = 0.001$, Fig. 6). Experimental enclosures also had minimal effects on skin microbiota. PP larvae of the transfer experiment did not differ from free-swimming pond larvae in community structure (pair-wise PERMANOVA: P(free)-PP: $t = 1.0523$, $P = 0.237$) or alpha diversity (Chao1: KW $\chi^2 = 0.79421$ $P = 0.3728$, PD: KW $\chi^2 = 1.7488$ $P = 0.186$) (Supplementary Fig. 3). SS larvae did, however, differ slightly in community structure (pair-wise PERMANOVA: S(free)-SS: $t = 1.2805$, $P = 0.02$). Phylogenetic diversity differed slightly between experimental larvae and free-swimming stream larvae ($df = 1$, KW $\chi^2 = 3.8977$ $P = 0.048$); but, this difference was not mirrored in Chao1 diversity ($df = 1$, KW $\chi^2 = 2.6599$ $P = 0.1029$) (Supplementary Fig. 3).

**Larval food intake and growth in habitat-switched larvae**. Out of 160 larvae at the start of the experiment, 148 survived. Two individuals from the habitat controls and 10 from the habitat-switched groups died before the end of the experiment. Performance of larvae was measured by growth rate. In general, pond larvae showed larger growth rates than stream larvae, which is associated with the higher water temperatures typically observed in ponds compared with spring-fed streams. Importantly, for the design of our experiment, the pertinent comparisons are the habitat-switched groups with their respective destination habitat-control groups, that is, PS compared with SS and SP compared with PP. Accordingly, growth rate of larvae in the PS group did not differ in comparison with that of the SS control group (Tukey-HSD: $P = 0.492$), and SP individuals exhibited a significantly higher growth rate than that of the PP control group (Tukey-HSD: $P = 0.001$, Fig. 7).

SP larvae preyed on copepods, ostracods and coleopteran larvae; PS larvae consumed ostracods, amphipods, dipteran larvae and coleopteran larvae. These taxa represent a broad range of the food spectrum typically found in pond and stream habitats as per the conducted field survey (field survey data are presented in Fig. 2). Furthermore, the proportion of individuals with food in

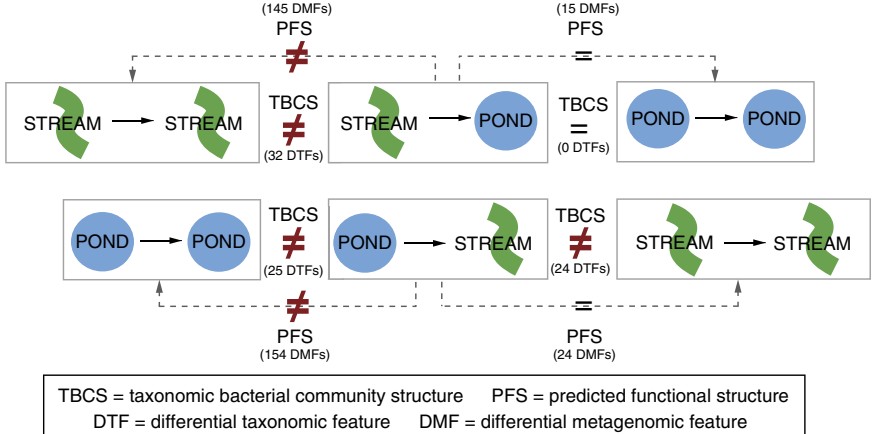

**Figure 5 | Schematic summary of gut microbiota results in response to habitat transfer of their hosts**. Importantly, despite differing taxonomic community structures between PS and SS groups, their predicted metagenomic features exhibit a lower number of differential features. For the LEfSe-based differential features, the total number of OTUs going into the analysis was 82, and the total number of metagenomic features was 259 metabolism-associated KEGG level 3 pathways.

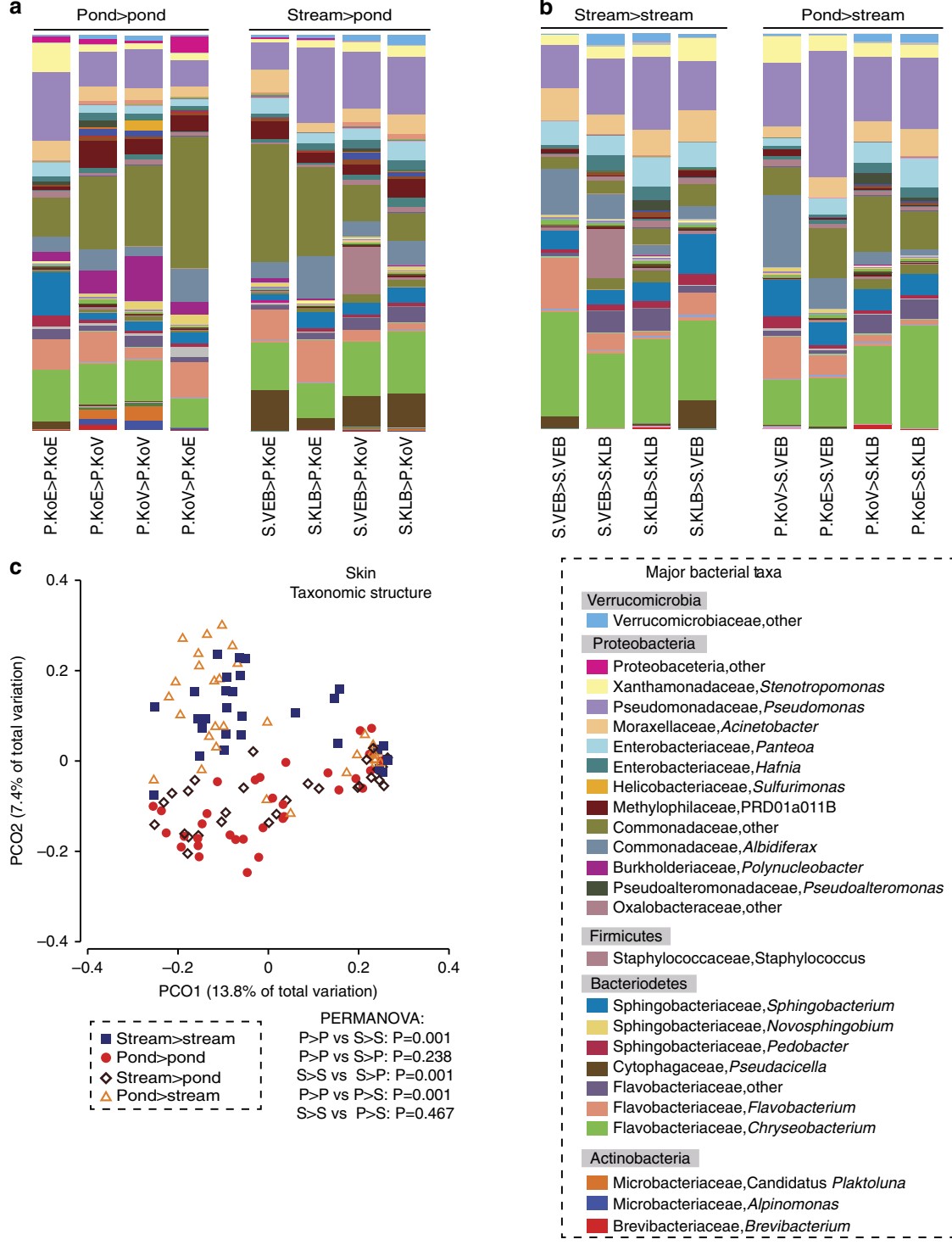

**Figure 6 | Skin bacterial community structure shifts uniformly in response to habitat transfer.** Habitat-switched individuals match the community structure of destination habitat controls. (**a**) (**b**) show mean relative abundance profiles of Core50 OTUs in the skin bacterial communities of larvae at the genus level. **a** highlights larvae derived from streams and transferred to ponds. **b** highlights larvae derived from ponds and transferred to streams. Major bacterial taxa are identified in the provided legend. **c** shows Principle Coordinate Analysis of unweighted Unifrac distance matrices of the full community for both habitat-switched group in comparison to both habitat-control groups. Sample sizes were: $n = 36$ (SS; blue squares), $n = 38$ (PP; red circles), $n = 27$ (SP; brown open squares), $n = 36$ (PS; yellow open triangles). Pair-wise PERMANOVA results are displayed below the PCoA. PERMDISP showed no differences in dispersions among groups ($F_{(3,118)} = 0.82197$, $P = 0.56$).

their stomachs did not differ between habitat-switched groups and their respective control groups (Fisher's Exact Test: $P > 0.05$), suggesting that larvae transferred into the other habitat type preyed and consumed habitat-specific food resources.

## Discussion

The factors dictating assembly and structure of complex host-associated microbial communities are of central interest in microbial ecology. The potential role of these communities in

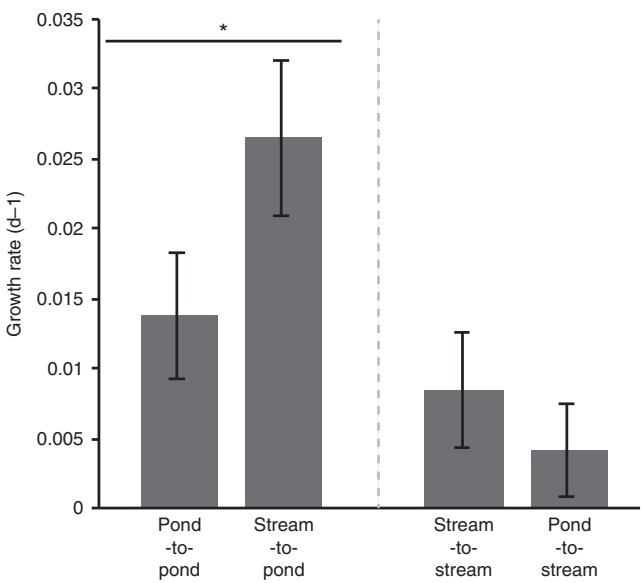

**Figure 7 | Mean growth rate for each group of larvae from the reciprocal transfer experiment.** Error bars represent 95% confidence levels, and asterisks denote statistically significant differences determined by ANOVA. Sample Sizes were: $n = 36$ (SS), $n = 38$ (PP), $n = 27$ (SP), $n = 36$ (PS).

facilitating vertebrate hosts' acclimation to new environments makes understanding them even more crucial in the face of rapid environmental change around the globe. In our study, we gained insight into the role of external environmental conditions in gut microbial community dynamics under fully natural conditions using reciprocal transfer between ecologically different habitats. Our data corroborate that environmental factors mediate gut bacterial community structure. We further demonstrate that functional potential can be met without taxonomic coherence and provide evidence to suggest the potential role of gut microbiota in mediating ecological adaptation.

Overall gut bacterial communities of salamander larvae were comprised of Proteobacteria, Firmicutes and Bacteriodetes (Supplementary Fig. 6), which mirrors the few other inventories on gut microbiota of amphibian larvae that have been published to date[25,48,49]. Snapshots of gut-associated bacterial communities obtained from the field survey revealed differences in bacterial diversity between pond and stream larvae as well as habitat-specific signatures in bacterial community composition and structure. Only 24% (20 of 82 OTUs) of the core50 OTUs were found to be associated with the core gut microbiota of both habitats. One of these OTUs (*Pseudomonas fluorescens* - EU774946.1.1391) was found to be differentially abundant between pond and stream larvae in the field survey and two OTUs were detected in the transfer experiment through LEfSe analysis (*Hafnia* sp.-AMQL01000001.12248.13793 and *Clostridium* sp.-EF590059.1.1364). In this study system, the habitat types strongly differ with respect to abiotic (e.g., temperature and oxygen content) as well as biotic conditions (food resources). The gut microbiome of stream larvae was enriched for Proteobacteria (aerobic bacteria), whereas pond larvae were enriched for Firmicutes (typically anaerobes). This may relate to the typically higher oxygen content of streams and the seasonal anaerobic conditions in ponds, suggesting the environment strongly influences the type of bacteria that can colonize the gut of the salamander larvae. Food spectrum analysis from larvae stomach contents confirmed that larvae from these contrasting habitats ingest different food items. Diet is known to influence gut microbial communities in fish[21,23], humans[19], other

mammals[20,50,51] and amphibians[25,48]. However, in many instances studies were unable to fully separate different dietary intake from host genetics (that is, host-specific factors within the gut environment)[23] because different species were compared. In our study, the factor of host species can be excluded given that pond and stream-type salamanders studied here represent recently diverged subpopulations (<8,000 years) of the same species living in sympatry in the same forest system[40]; therefore, these data explicitly show that environmental conditions and diet can affect gut bacterial community structure under fully natural conditions. Furthermore, predicted functions of these gut bacterial communities differed, which is likely associated with the intake and digestion of different food substrates. Host organisms in many cases rely on gut-associated microbiota to degrade complex substrates into nutrients usable by the host[5,6]. The different food sources ingested between pond and stream larvae likely induced bacteria with different suites of enzymatic activities to promote host digestion. Differential predicted functional features, such as increased lipid metabolism in stream larvae and increased carbohydrate metabolism pathways (that is, starch and sucrose metabolism and fructose and mannose metabolism) in pond larvae, would support such a hypothesis.

With the reciprocal transfer experiment, we further confirmed the role of the environment in shaping gut bacterial community structure and investigated how gut microbiota may respond to environmental change. Given the importance of abiotic and biotic environmental conditions in shaping gut microbiota from previous studies comparing different hosts, it was predicted that individuals transferred into the non-native habitat (that is, habitat-switched individuals) would shift and exhibit gut bacterial communities similar to individuals naturally found there. However, we observed differential patterns in the gut microbiota depending on the directionality of the transfer (see Figs 4 and 5). Strikingly, despite these different patterns from the structural perspective, predicted functions of habitat-switched larvae matched those of the destination habitat-control larvae in both cases. This result, however, needs confirmation by direct metagenomic and metatranscriptomic sequencing. The response of the skin microbial communities of the same experimental individuals further highlights the unique nature of the differential response seen in the gut microbiota. Here, the shift in community structure to that of larvae from the destination habitat was observed in both habitat-switched groups. Thus, the composition of gut microbiota, regulated both by external abiotic as well as biotic (mainly diet) environmental parameters, appears to respond distinctively in comparison with the skin, which is mainly influenced by abiotic factors.

The divergent taxonomic result between habitat-switched groups but equivalent matching of the predicted functions to that of the destination habitat for the gut bacterial communities demonstrates that gut bacterial communities of these larvae may be displaying alternative responses to the new environment; one being a complete turnover to the community of native larvae (that is, SP larvae), and the second being an alternative stable state[52] that still allows them to match predicted functions of a new environment (that is, PS larvae). This functional convergence in the PS larvae could be a result of functions brought by a limited number of new members recruited into the community or changes in relative abundances of existing taxa, and suggests that function can be matched without taxonomic coherence. Owing to the predictive nature of PICRUSt and its use here with a non-model host, metagenomic or transcriptomic sequencing will be an important future research direction for verifying our results related to gut microbial function.

The shift in gut community structure of SP larvae to match that of PP controls was the expected result. Upon entry into the pond

environment, stream larvae are met not only with a new microbial reservoir that harbours new potential colonizers but also with new food resources that must be consumed and effectively digested. Such changes can lead to 'new' bacteria replacing existing gut members perhaps because they are competitively dominant in the new abiotic conditions as well as in utilising new available nutrients. But, what might drive the unexpected taxonomic pattern seen in the PS group? Colonisation history and priority effects can have a strong influence on community succession[2,12,18,47]; therefore, the existing gut communities in these larvae likely have a role in the observed pattern. Perhaps the lower bacterial diversity of pond larvae prohibits community conversion to that of the SS controls. A similar finding was observed for mice, where individuals fed with low-fibre diets consistently through generations had reduced gut bacterial diversity and these communities were 'incapable' of reverting back to high-fibre type gut microbiota even when fed such a diet[53]. Alternatively, the concept of microbial community resistance may be at play; in ecological theory, the concept of resistance characterizes communities that remain essentially unchanged despite disturbance[54–56]. Such resistance can be facilitated by host immune regulation[57] but also by the microbial community itself[55,58]. If a microbial community contains members that are physiologically versatile or plastic, then its composition may be more resistant to disturbance[55,59]. The gut bacterial community structure of PS larvae could be considered more resistant in that the taxonomic composition experiences less divergence from the gut community structure of pond-habitat controls. Therefore, ecological concepts related to community resistance could, in part, explain the observed pattern in gut communities of PS larvae.

Gut bacterial communities of pond larvae may be more resistant because they have many members that are ecologically versatile and can dwell under different abiotic and biotic conditions. In such a case, these bacteria could still be competitive and successful if the environment changes. At the functional level, such versatile taxa could also exhibit metabolic plasticity[60], thereby changing their functional output in response to the new environment. In addition, these communities could contain many members able to undergo dormancy, that is, a state of reduced metabolic activity[61], which remain detectable through our DNA-based assessment but are not functionally active in the new environment. As PICRUSt can only predict functions based on 16S marker genes, we cannot discern if such hypotheses are at play, however, both are plausible. Shotgun transcriptomics of microbial mRNA or targeted sequencing of transcribed 16S rRNA could elucidate such hypotheses and would be an important direction for future research. In either case, such flexibility could be expected in gut bacterial communities of pond larvae, which experience severe and fast fluctuations in temperature and water level, as well as food composition and availability[44]. Such 'pulse disturbance' certainly would favour versatile and plastic microbes and might favour bacteria able to undergo dormancy[55]. On the contrary, the stream environment has more or less optimal and constant resources; and therefore, bacterial communities, which do not require the ability to respond to or accommodate fluctuating environmental conditions, can exist[55] and may be more easily replaced when the host is transferred into a new environment.

Evidence increasingly supports the idea that host microbiota has a role in vertebrate phenomic plasticity (that is, the capacity of a single genotype to change its expression so as to exhibit different phenotypes in response to environmental pressures[12]), and may be influencing vertebrate host evolution[12,62,63]. The capacity of a host's gut microbiota to change its composition (for example, the gain and loss of taxa as well as shifts in relative abundance) or gene expression in response to physiological changes in the host or external environmental changes has been termed 'metagenomic plasticity'[12]. In the context of the hologenome concept, a vertebrates' acclimation to novel environments could be driven not only by interaction of the host genome with the environment, but by the interaction of their hologenome (that is, the cumulative genomes of the host and its microbial symbionts) with the environment[12,64,65]. Therefore, metagenomic plasticity of host microbial symbionts may boost the ability of hosts to acclimate and adapt[12]. In this framework, our results on growth rate provide evidence that microbial rearrangements of the gut microbiota may benefit the host when faced with new environmental conditions. Salamander larvae were able to cope with novel habitat conditions, with no fitness consequences (for example, no obvious negative performance with respect to growth rate (Fig. 7)). This could be a result of the rapid capacity of the gut microbiota of these larvae to functionally switch, albeit by different taxonomic means, thereby boosting the hosts' acclimation capacity to the new environment. Future studies, perhaps through control experiments with germ-free larvae, will be important for teasing apart the potential benefits of host microbiota in facilitating host acclimation to changing environments. In a broader context, this potential 'metagenomic plasticity'[12] could be facilitating the ecological adaptation and divergence of these salamanders between pond and stream habitats.

## Methods

**Study species and system.** Fire salamanders (*S. salamandra*) are a unique, non-model, vertebrate organism. Though strictly terrestrial as adults, females of this species deposit fully developed larvae in aquatic habitats, where larvae develop until metamorphosis. In the Kottenforst (near Bonn, Germany) the adaptation to deposit larvae into different habitat types, that is, small first order streams versus stagnant ephemeral water bodies (for example, ponds, tire ruts and ditches; hereafter called ponds) has caused recent adaptive divergence as well as behavioural differentiation within the salamander population[39–42]. The streams and ponds in which larvae develop differ in many abiotic as well as biotic conditions. Although streams represent a stable environment with low temperatures and constant water supply throughout the year, ponds are less predictable with high variation in temperature and a high risk of rapid desiccation[43]. Both habitat types also differ in the composition of potential and preyed upon food resources (Fig. 2). Furthermore, restriction of food availability in ponds (that is, starvation) has resulted in habitat-specific adaptations, such as cannibalism and a shorter time to metamorphosis[39,44].

**Field survey.** The habitat-specificity of food spectra and gut microbial community structure of larvae from stream- and pond-habitat types was assessed by carrying out a field survey on larvae from representative ponds and streams across the Kottenforst. Permission was received by the Nature Reserve Authority of the city of Bonn and the Ethical Committee of the Technical University of Braunschweig for sampling salamanders in the Kottenforst for both the field survey and transfer experiment (see below). Fire salamander (*S. salamandra*) larvae were freshly caught by dip netting during spring 2015 (28th and 29th April) from three pond sites (KoV, KoE and KoK) and four stream sites (Vennerbach, VEB; Keltersbaumbach KBB; Annaberger Bach, ANB; Hittelbach, HIB). These larvae are easily distinguishable from other amphibian larvae that may co-occur. Altogether, 104 (60 stream and 42 pond) larvae were collected, transported to the field station/laboratory live, in individual containers and immediately (within 5 h of collection) euthanized with Tricaine Methanesulfonate (MS222). Subsequently, larvae were dissected and the stomach and intestine portion of the digestive tract were separated. Stomachs were preserved in 80% ethanol for visual food spectrum analysis, and intestines were frozen at −80 °C immediately after dissection and stored until DNA extraction for gut bacterial community analysis.

**Reciprocal transfer experiment between ponds and streams.** To analyse the response of gut microbial communities to environmental change, a reciprocal transfer experiment was performed with salamander larvae between the pond and stream habitats. This experiment had a fully factorial design using two replicate sites for each habitat type. Two pond sites (KoV and KoE) and two stream sites (VEB; Klufterbach KLB) were used. The experiment was set up such that 30 larvae from each site were transferred to the three other sites, with each site receiving 10 larvae (total $n = 160$). In addition, 10 larvae remained in their original site (see Fig. 3 for details). Individuals that switched habitat type, that is PS and

SP individuals, were considered 'habitat-switched individuals', whereas individuals that remained in their native habitat, that is PP and SS individuals, were considered 'control individuals'. Each larva was housed separately in a single semipermeable container (HDPE plastic, L8.65 × W8.65 × H17.85 cm plus screw-cap) equipped with two circular stainless-steel grid windows (5.0 cm diameter, mesh size 3.15 × 3.15 mm) and Styrofoam floaters (allowing containers to remain near the surface for oxygen exchange; Supplementary Fig. 6). Mesh size of the grid windows allowed possible food items to enter, while also preventing escape of the sala-mander larvae. Containers were placed near the pond edges close to the surface, and in the bed of streams, which is where larvae naturally spend their time. For each site, container placement was randomized with respect to larvae origin site. The experiment started on 11 May 2015 and ended 26 May, that is, a two-week duration. We used larvae in an early developmental stage to avoid metamorphosis occurring during the transfer experiment; no individuals underwent metamorphosis during the experiment. The weight of each larva was measured at the start and end of the experiment. At the end of the experiment, larvae were removed from their containers and processed in the same manner as described for larvae in the field survey, with one addition; skin microbial communities were sampled on each individual prior to euthanization. Each larva was handled with a clean set of gloves to avoid cross contamination between larvae, and rinsed with 30 ml of sterile water. After rinsing, each larva was swabbed across the whole body with 10 strokes and the swab samples were frozen at −80 °C until DNA extraction for bacterial community analysis.

**Food spectrum analysis.** Visual food spectrum analysis was performed following Reinhardt et al.[43] for both field survey and transfer experiment individuals. Only the stomach proportion of the digestive tract was considered and all food items within the stomach were examined. All food organisms were classified into broad taxonomic groups, mainly corresponding to order or genus level taxonomy.

**DNA extraction.** Genomic DNA was extracted from larvae intestines and skin swabs using the MoBio PowerSoil-htp 96-well DNA Isolation kit (MoBio Valencia, CA) with minor adjustments to the manufacturer's protocol to increase DNA yield, including a 10 min incubation at 65 °C after C1 addition and 10 min incubation at room temperature after C6 addition to the spin column. In addition, centrifuge times were doubled to account for slower rotor speed. DNA extracts were stored at −20 °C until further processing.

The V4 region of the bacterial 16S rRNA gene was PCR-amplified with the 515F and 806R barcoded-primers using a dual-index approach[66]. PCR-reactions contained: 12.4 μl of DNA-free water, 4 μl HF Buffer (High Fidelity PCR Buffer, New England Biolabs (Ipwich, MA)), 0.2 μl of Phusion Taq, 0.4 μl of dNTPs (10 μM), 0.5 μl of each primer (10 μM) and 2.0 μl of sample DNA. PCR-conditions were: a denaturation step of 98 °C for 20 s, followed by 28 cycles at 98 °C for 10 s, 55 °C for 30 s and 72 °C for 30 s, and a final extension at 72 °C for 10 min. Each sample was amplified in duplicate PCR-reactions and then the PCR-products were combined. Combined samples were pooled together in approximately equal concentration (as determined by gel band strength), and then cleaned used the Qiagen MiniElute Gel Extraction Kit (Qiagen, Germantown USA). The final DNA concentration was determined on a Qubit fluorometer using a broad-range dsDNA kit. Sequencing was performed using paired-end v2 chemistry on an Illumina MiSeq at the Helmholtz Center for Infection Biology (Braunschweig, Germany).

**Sequence processing.** Quantitative Insights Into Microbial Ecology (MacQIIME v1.9.1) was used for all sequence processing unless otherwise stated[67]. In brief, paired-end sequences were merged with fastq-join, quality filtered (using QIIME defaults), and filtered by sequence length to include only those between 250 and 253 basepairs (usegalaxy.org). Usearch61 de novo based chimera detection within QIIME was used to identify chimeras on a per sample basis (http://drive5.com/usearch/usearch_docs.html)[68], and the identified chimeric sequences were removed from the quality-filtered sequences prior to OTU-picking. Sequences were clustered into OTUs at 97% similarity using an open-reference OTU-picking strategy[69]; http://qiime.org/tutorials/open_reference_illumina_processing.html). The SILVA 119 release (24 July 2014; https://www.arbsilva.de) was used as the reference database, and the UCLUST[68] algorithm was used in the de novo clustering step. The most abundant sequence from each OTU was selected as a representative sequence and these representative sequences were aligned using PyNAST[70]. The RDP classifier[71] was used to assign taxonomy with the SILVA 119 'majority' taxonomy as the reference. A phylogenetic tree was built using FastTree[72] adhering to QIIME's standard procedures. OTUs with <0.001% of the total reads of all samples together were removed as recommended in Bokulich et al.[73], yielding a total of 1,624,632 sequences (range: 1003–51,579 sequences/samples; average: 6170.4). All samples were rarefed at 1,000 reads to allow inclusion of a large portion of the samples and capture the majority of the diversity present within larva gut and skin bacterial communities (Supplementary Fig. 7). Samples with below 1,000 reads were therefore excluded from analysis. After sequence filtering and rarefaction, 77 (46 stream larvae and 31 pond larvae) out of 104 samples remained in the field survey dataset, and 137 (36 SS, 38 PP, 27 SP, 36 PS) out of 148 samples (total number of individuals that survived the experiment) for each the gut and skin data sets remained in the transfer experiment dataset.

**Analysis of bacterial communities and predicted functions.** Alpha and beta diversity were calculated in QIIME, and PERMANOVA and Principle Coordinate Analyses (PCoA) were completed in Primer7[74] in all cases. LEfSe analyses[75] and PICRUSt predictions[76] were completed using the Galaxy platform (http://huttenhower.sph.harvard.edu/galaxy/). All other statistical analyses were completed in R (Version 3.2.3).

The habitat-specificity of salamander larva gut bacterial communities was evaluated by examining the field survey data set. Alpha diversity was calculated using Chao1, Faith's phylogenetic diversity, and Simpson's evenness indices, and these indices were compared with Kruskal–Wallis tests. Beta diversity was calculated with unweighted UniFrac, weighted UniFrac and Bray-Curtis metrics, and PERMANOVAs were performed to determine whether there were differences between larval habitat (pond or stream). To explore the bacterial taxa that may be responsible for the observed differences in community structure a LEfSe was performed on the Core50 community. The Core50 gut community was calculated as follows: the OTU table was filtered to include only the OTUs present in a minimum of 50% of individuals from either ponds or streams. This calculation was performed separately for pond (gut Core50: n = 30 OTUs) and stream (gut Core50: n = 72 OTUs) samples and subsequently compiled to avoid excluding OTUs that may be specific to only one habitat type. In combination, the gut Core50 contained 82 unique OTUs. LEfSe analysis was performed on the Core50 in order to focus the analysis on the most prevalent and abundant OTUs and to minimize the number of comparisons performed, therefore reducing Type 1 error. Habitat type (pond/stream) was used as the class variable and site (KoV, KoE, KoK for pond and VEB, KBB, ANB, HIB for stream) was used as the subclass variable. The defaults parameters were used in all completed LEfSe analyses.

To examine whether pond and stream larva bacterial communities exhibited different predicted functions, the bioinformatics tool PICRUSt was used to predict the metagenome of each sample. Importantly, metagenome predictions depend on whether the bacterial taxa present within the samples are represented in the genome database. The NSTI (Nearest Sequenced Taxon Index) measures this relationship, with lower values illustrating a closer mean relationship[76]. The gut bacterial community samples of salamander larvae had an average NSTI value of 0.047 ± SE 0.001, which indicates good coverage. It is important to stress that we used PICRUSt predictions in a comparative context to illustrate specific predicted functional differences between groups and not with the goal of describing the metagenome potential of these communities. PICRUSt[76] was performed on the Core50 OTU table to focus the predicted bacterial function analysis on the most prevalent and abundant OTUs. First, OTUs were assigned a Greengenes 13.5 (May 2013 release[77], OTU identity using the closed reference OTU-picking strategy, because this taxonomy is necessary for using PICRUSt. Copy number normalisation of each OTU, metagenome prediction of each sample and functional categorisation based on the KEGG Orthology were performed with PICRUSt on the Galaxy platform (https://huttenhower.sph.harvard.edu/galaxy/), and these steps generated a table with the KEGG pathway abundances for each sample. Pathways with <10 counts were removed from the table. This table was subsequently rarified to an even sampling depth of 100,000 counts. Beta diversity was calculated using the Bray–Curtis metric and PERMANOVA and PCo Analysis were used to compare and visualize predicted functional profiles. In addition, LEfSe was used to determine which metabolism-associated functional features (if any) were differentially abundant between pond and stream larvae.

For the transfer experiment, all analyses on the gut microbial communities were completed with the Core50 OTUs from the field survey. This allowed for a focused look at OTUs that are common, and in most cases, abundant members. Beta diversity was calculated as explained above. A single PERMANOVA was completed to test for differences in bacterial community composition and structure among experimental treatments (SS, SP, PP, PS), and subsequent pair-wise treatment comparisons were completed to further explore the main effects. PCo Analyses were completed to visualize beta diversity among experimental groups. Given the hypothesis that gut bacterial communities of habitat-switched individuals may experience a shift in OTUs from their origin habitat control, LEfSe analyses were performed between each habitat-switched group and its origin habitat-control group (for example, PS was tested against PP). LEfSe analyses were completed on the Core50 gut community; experimental treatment was used as class variable and origin site name was used as subclass variable.

The skin microbial community samples from the transfer experiment were analysed in a similar framework with the exception of how the Core50 OTUs were calculated. Here, Core50 was calculated as follows: the OTU table was filtered to include only the OTUs present on a minimum of 50% of individuals from either pond or stream larvae free-living at the time of experimental sampling (that is, the end of the reciprocal transfer experiment). This calculation was performed separately for each habitat and subsequently compiled to avoid excluding OTUs that may be specific to only one group. For the skin Core50, stream individuals had 60 OTUs and pond individuals had 84 OTUs, which totalled 105 unique OTUs. Beta diversity, PERMANOVAs and PCo Analyses were completed as explained above.

To explore the relationship among predicted functions of each habitat-switched group and habitat-control groups the metagenome of each sample was predicted with PICRUSt as explained for the field survey. PCo Analysis and a PERMANOVA were completed to visualize and compare these predicted metagenomic profiles, and pair-wise LEfSe analyses of each habitat-switched group to each habitat-

control group were executed to identify differential functional features. For LEfSe, experimental treatment was the class variable and origin site name was the subclass variable.

**Analysis of larval performance.** In order to provide a rough estimate of larvae performance of the treatment groups, mean growth rates were determined from individual fresh weight with a logistic growth equation. Growth rates $(g\,d^{-1})$ for the sampling interval were calculated with the following equation: $G = (\ln(W_{t+1}) - \ln(W_t))/t$, where, $W_t$ is mean larval fresh weight at the start, $W_{t+1}$ is fresh weight at the end of the experiment, and $t$ indicates the time period (days) between the start and end of the experiment. One-way analysis of variance with subsequent Tukey *post hoc* tests were completed to compare growth rate between each habitat-switched group and the respective controls.

**Data availability.** Sequence data have been deposited in the Sequence Read Database (SRA) under project ID SRP074716 (BioProject PRJNA320968). All other data are available upon request from the authors.

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

## Acknowledgements

We are grateful to Laura Duntsch for her assistance during fieldwork. This study has been supported by grants of the German Research Foundation (DFG) provided to SS (STE1130/8-1) and MV (VE247/9-1). We also want to thank the Amt für Umwelt, Verbraucherschutz und Lokale Agenda of the city of Bonn for granting research and sampling permits for salamanders in the Kottenforst.

## Author contributions

S.S. and D.G. designed the conducted experiments. D.G., E.S. and T.R. conducted field surveys and sample collection for the experiments. M.C.B. performed laboratory work and completed sequence analysis. S.B., R.G. and M.J. performed sequencing. T.R. completed food spectrum analysis. M.C.B. wrote the manuscript with substantial input from S.S., and M.V. and C.C.T.

## Additional information

**Competing financial interests:** The authors declare no competing financial interests.

