## [Peer Review File · Nature Communications]

Reviewers' comments:

Reviewer #1 (Remarks to the Author):

The original study by Bletz et al., "Gut microbiota of amphibian larvae shift differentially in community structure but converge on habitat-specific predicted functions", is an excellent contribution to microbial ecology with well-designed field experiments, and state-of-the-art analyses. The authors are able to demonstrate with a reciprocal transplant experiment that gut microbiota either shifts to match the microbial functions of destination habitat and diet, and this shift can occur either by a shift to match destination community composition, or a shift in community composition that doesn't match the destination, but still shifts to match the functional profile of the destination. Thus, functional profile is not tied to specific microbial composition, and the authors suggest that historical contingency and colonization resistance can explain incomplete community composition shifts. They also contrast gut community shifts with skin community shifts. The abundance of data generated by this study is clearly and well presented.

Specific comments and room for improvements:

Abstract line 39 delete "fully" and is there space to mention that skin communities were also measured for comparison with gut communities?

Line 51 change to "Host factors such as stomach pH, mucuins,..."

Line 82 There's no such thing as 'rDNA', don't use this term

85 change to "understanding exists of the external environment's influence of these communities and of how..."

87 change to "...organisms. Currently, we are lacking ..." this sentence should be re-worded, we will probably never have a "complete" understanding.

110 delete "as a valuable tool"

112 delete "This community..." and reword "Amphibian microbiota are well studied, and known for defensive ..."

114 change to "considered to be more..."

116 delete "abiotic and biotic" change to "which have additional nutritional (i.e. diet) influences." Stress and the immune system are biotic influences on both skin and gut body sites.

199 change to "larvae, gut..."

123 and everywhere else delete "complete". I don't think you show complete matching, but rather not significantly different?

162 Here summarize the differences in function that are described in the discussion.

Section starting line 179 – here it is not clear why there would be any changes in pond-to-pond transplants or in stream-stream transplants. Two kinds of analyses would help explain this, or

provide interesting additional results: 1. Describe in the text the results from fig. 4 showing little difference in stream to “native stream” vs. stream to “non-native” stream. Same for ponds. 2. What were the changes from field to semi-captive experimental conditions? Did diversity decrease after putting larvae in enclosures? Do enclosures/different food access explain these small pond-pond and stream-stream differences? This could be mentioned in text with a fuller analysis in the supplement.

231 change “naturally” to “natural”

249 change to “intake and digestion of different...” Microbiota are not changing because the host “needs” them to.

252 delete “required” and change to “likely induced bacteria with different suites of enzymatic activities, and promoted host digestion.”

255 Some of these metabolism differences should be in the results section also.

280, also 288 and “full” in 291 – Why would you expect a complete shift?

Define complete. Do you mean that a shift occurred so that the transplant community was not significantly different from the control community?

281 change to “to that of larvae”

295 delete “flora” since you are not studying plants. “biota” would be appropriate.

302 How does this compare with findings of Kueng et al. 2014 Stability of microbiota facilitated by host immune regulation: informing probiotic strategies to manage amphibian disease. PlosOne, where adult amphibians showed skin colonization resistance to disturbance?

315. Why only shotgun? Wouldn't targeted 16S rRNA transcriptomics also work to get at dormancy?

I suggest a table or revised fig 5 that includes % OTUs that significantly change.

367 change board to “broad”

Figs 2 and 4 – the letters on figures do not match the legend text.

Fig 4 legend. 644 Do you mean “identical” or not sig. different?

When discussing, I would expect that salamanders have a core set of bacterial OTUs that stay constant among all conditions and define the species microbiome. Species identity is thought to be a stronger predictor or driver of microbiota than environmental conditions, at least for the skin. Can you comment on this? What percent of OTUs differed between habitats?

Fig. 5 I would like to see, in addition to features, the % difference in terms of proportion of OTUs based on the LefSe analyses. An additional part to this figure, or a supplement could

indicate the same things in the gut and skin for field pond to enclosure pond, and field stream to enclosure stream. There are some interesting changes occurring, just by putting the larvae into the enclosures. Are there enclosure specific OTUs?

Reviewer #2 (Remarks to the Author):

In this study, Bletz et al. analyze the effects of the ecological niche on the structure, composition and function of gut microbiomes in an interesting animal (amphibian) model – Salamandra salamandra. The authors designed an ecological transfer experiment, where pond-adapted salamanders were transferred to a stream environment, and vice versa. This interesting experimental design allowed the authors to measure the impact of both abiotic and biotic parameters of the ecological niche on the gut microbiomes.

Overall, I have found this paper a stimulating read, with an interesting model (the two salamander populations have recently undergone ecological differentiation), a well-conceived experimental setup and stimulating biological conclusions. The paper will be of interest to the community of microbial ecologists.

However, I think that many parts of the manuscript need to be improved or clarified, and some of the statements need to be toned down.

-- Main concerns

1. The authors are not measuring the functional profiles of these microbiomes with metagenomics sequencing. Instead, they use 16S and an in-silico program to predict functions based on the taxonomic profiles of these communities. To me this is fine as long as the authors admit and discuss the limits of the approach. Instead, they attempt to convince the reader that it is better than doing metagenomics to characterize functional profiles. Even though metagenomics can “overlook functions from less abundant members” (line 76), it is also the case with 16S. 16S amplicon sequencing also depends on pseudo-universal primers that actually miss a large component of the microbial biodiversity, which has been shown in many studies.

Metagenomics is much less affected by this issue.

Furthermore, it is stated that host DNA could negatively impact the sequencing of microbial communities and subsequent analyses (line 76-78). This is very controversial, and there is no general rule for this. The reference cited at this stage to support this claim is not convincing in this context.

Still around this issue, the PICRUSt program has its own limits too, especially when applied to

non-human hosts. As it predicts functions, there is uncertainty on these predictions, which should also be acknowledged. Langille et al. showed in their original paper that PICRUSt is very accurate at predicting functions for human microbiomes, but that it is much less accurate to do so in other non-human mammals. What stands for non-human mammals is also very likely to stand for amphibians, as the reference genomes used by PICRUSt are biased towards human-associated microbes.

Overall, I don't think the authors should be criticized for not having performed metagenomics, especially because their conclusions on functional profiles make sense regarding the original hypotheses. But I think the introduction needs to be rephrased and that conclusions drawn from PICRUSt experiments should be discussed to incorporate these remarks.

2. To what extent the gut microbiome of salamander larvae reflects the microbiome of its aquatic environment? One potential issue here is whether the authors are observing profound changes in the microbiome because new bacteria are colonizing per se the gut after the transfer to a pond or a stream, or whether the bacteria are transiently passing through and are actually living in the water. The skin microbiome experiment can control for this to some extent, but the skin in amphibian is also known to host a specific microbial community that is not entirely influenced by the surrounding water. The authors should discuss this.

3. The authors name their experiment a "reciprocal transfer" at line 107, which I think is good. But then in the rest of the manuscript, the terms "transplant", "transplanted" and "transplantation" are used. To me this is confusion with the gut microbiome literature, in which the word "transplant" is used at length to talk about microbiome transplant (such as Fecal Microbiota Transplantation (FMT)), which is also mentioned by the author at line 53. When reading the manuscript, it took me time to realize that the author are transferring or transplanting individuals into different environments, and not transplanting bacterial communities themselves. I think the word "transfer" should be preferred in this manuscript to avoid confusion, especially in the context of this literature.

4. There are many issues with the figures and their legends:

- First, Figure 1 is cited in the text after Figure 2 and 3. As Figure 1 presents the ecological transfer experiment, it should be placed as figure 3.
- Line 206: Figure 7 is cited, but this is Figure 6.
- In Figure 1 legend: please explain what P.KoE, P.KoV etc means. Even if these acronyms are explained in the Methods, it would help the reader to have this info right below the figure to more easily understand the experiment.
- Figure 2B: In this PCoA plot, the two axes are "PCO1". I guess the y axis represents the second component of the PCoA?

- Figure 2 legend: A) and B) do not point to the right plots. A) should be the alpha diversity and B) should be the PCoA.

- Line 625: analyses -> analysis

5. This paper is also interesting in the context of biological adaptations promoted by gut microbiome plasticity. Few studies have designed experiments to measure the benefit that gut microbiome rearrangements provide to the host when the host faces new environmental conditions. In this context, the measurement of growth rate following the ecological transfers is interesting. The higher growth rate observed for the stream-to-pond individuals is intriguing, because it corresponds to the higher growth rate observed among individuals living in pond environments. However, it is difficult to conclude whether the microbiome rearrangements following the stream-to-pond transfers is causing this higher growth rate, or whether it is simply the effect of the higher temperature in pond waters. An ideal solution would be to obtain germ-free larvae, and compare the growth rate of larvae living in hot (pond) vs. cold (stream) temperatures, and measure the benefit of the microbiome with respect to this baseline difference. Maybe the authors would be interested in discussing these topics in their Discussion section? (Especially because at the moment the discussion unnecessarily reiterates some of the main conclusions stated in the Results).

Importantly, several papers that are relevant to this study are not cited and the results should be discussed in the context of these papers, to feed the discussion regarding the impact of environmental change on the gut microbiome:

- Alberdi, A. et al. (2016) Trends in Ecology and Evolution.

<http://dx.doi.org/10.1016/j.tree.2016.06.008>

- Sommer, F. et al. (2016) Cell Rep. 14, 1655–1661

- Chevalier, C. et al. (2015) Cell 163, 1360–1374

6. The authors find that pond-to-stream larvae converge in functions but not in taxonomy with stream larvae. This is interesting, but the authors do not explain how this happens. They propose different hypotheses in the Discussion (line 276-280), but it is not obvious to me why they cannot test them. What are the different taxa that are functionally redundant? Are they phylogenetically distantly related?

Importantly, is the conclusion that taxonomic convergence is not found dependent on the similarity threshold that is used to define bacterial entities (97%)? Can convergence be seen at higher taxonomic levels? (lower similarity cutoffs, say 95, or 93%?). If yes, it would then indicate that taxonomic signatures of pond vs. stream are simply determined at different taxonomic levels along the bacterial tree, and that searching for convergence at 97% for stream

larvae is bound to produce divergent signatures (while still producing convergent functions).

-- Minor concerns

Line 72: In addition, -> In addition

Line 88: “complete” lack of understanding is too strong.

Line 147: As this is the first time you mention “LEfSe LDA” in the manuscript, please spell it out so that the reader immediately understand what you did.

Line 156-162: this paragraph is a little weak with no clear conclusion. Please expand with more details and impact.

Line 279: I don't understand what this sentence means.

Reference 10: animalcules -> animalicules

Signed: Mathieu Groussin

Reviewer #3 (Remarks to the Author):

To address the influence of environmental factors (both diet as well as abiotic factors) in the development of host-associated microbial communities, the authors analyze gut and skin communities from salamander larvae collected at two different aquatic habitats: ponds or streams, and those from larvae transplanted to the other habitat. In addition to a 16S rRNA analysis, they used PICRUSt to infer functionalities of the microbial communities.

There has been very few papers on the salamander microbiome, in particular that of the gut, so this paper is novel and will provide much needed data for this understudied group.

Most graphs are well made, and text and experimental set up are for the most part clear. I have mainly minor comments.

General comments

1. The Introduction felt very long – the first 2 paragraphs could maybe be summarized in 2-3 sentences about the essential role of the gut microbiota and possible factors that influence its

composition.

2. How do the communities found in/on salamander gut and skin compare to previously published data in salamanders or frogs? The authors could first point out that not many studies have been performed on this animal group, and that this study fills that gap, even without performing the transplant analysis. The authors could then compare their skin microbiota data to that published in e.g. doi: 10.3354/dao02004 and 10.1038/ismej.2007.110 (both not by this reviewer). Do multiple studies find the same bacterial groups?

3. How many of the 16S rRNA gene OTUs were new, i.e. not seen before in previous studies? Given the sparse data on salamander microbiota, I would expect that this study has found many novel taxa. Could the authors do e.g. a BLAST search and report the percent of OTUs that are <97% identical to known sequences? Alternatively, they could use the percent of OTUs that did not have a >97% match in the reference database during OTU binning. This would be a nice datapoint to add to this study.

4. PICRUSt. Maybe the authors could add some wording to express caution about the use of PICRUSt in this particular setting. It performs well in case of well-characterized and genome-sequenced microbial species, but how well would it perform with microbial species from a less well-studied sample type such as salamander gut? PICRUSt relies heavily on the presence of sequenced genomes, and I am not sure if it would perform very well in a situation where not many microbial members of a community have been sequenced.

5. Where all larvae from the same salamander species? Could the larvae belong to multiple species or was species determination easy to do? Could the species (mentioned in the Introduction) also be added to the Methods?

6. Could the higher "resistance" to change of the pond-to-stream group be explained by the lower species richness/diversity of the pond communities as shown in Figure 2? This might be an obvious, yet overlooked explanation (maybe similar to extinction of microbes found in subsequent generations when given a low-fiber, and then a high-fiber diet – once gone, they cannot revert back, Sonnenburg, doi:10.1038/nature16504, not by this reviewer). Maybe the authors could add this thought to the end of the Discussion.

7. No accession number has yet been provided, so this review could not evaluate these.

Specific comments

8. L47. I'm not a big fan of the "forgotten organ" term – microbiologists have long recognized that gut microbiota plays an important role in digestion, and it is not an organ either.

9. L58. "select the composition of the microbial community" or "influence the composition..." would be better here.

10. L60. Although I agree that temperature could be considered an abiotic environmental condition, it could also be a host-associated factor (such as those listed in L51), at least in e.g. mammals and birds. This text felt very long and split up into not-very-clearly defined groups.

11. L66. Can you really say that a host "acquires" a new habitat?

12. L71 and other places: Please use "16S", with a capital S (Svedberg).

13. L72. "In addition to the analysis of the structural response, it is perhaps even more important

to study whether such..." would be better.

14. L82. This should read "16S rRNA gene".

15. L102. Please use "16S rRNA gene amplicon"

16. L109. "We utilized..." sounded similar to what was already stated in L104-107.

17. L112. To what does "this community" refer? To the communities of salamanders or to skin communities? Please clarify.

18. L118. The last part of the Introduction, starting with "We found..." could maybe be deleted here and integrated into the Discussion.

19. L121. Why was this unexpected? The previous text in the Introduction suggested that it would change. The shift itself was not unexpected, because communities were expected to shift, but maybe the authors meant that the direction/nature of the shift was unexpected?

20. L129. Please use "16S rRNA gene amplicon".

21. L135-138. Since Figure 2B only shows one of these three distance metrics, this should probably be worded differently. It currently sounds as if 2B will show all three analyses.

22. L172. The current placement of the Figure 1 referral suggests that Figure 1 shows the responses. Suggestion to change text to: "reciprocal transplant experiment (Fig. 1) showed distinct responses".

23. L204. Would "shifted completely to match that of the destination habitat control individuals" be more clear?

24. L206. Should this be Figure 6?

25. L327. Please add more information about the species of salamander that was used in this study. Is this an endangered species? Did the authors need any permits or other permission to catch salamanders? Also, where these larvae collected around the same time as the larvae for the transplant experiment (i.e., May 2015)?

26. L329. Please add the location where the Kottenforst is situated (e.g. near the City of Bonn, Germany).

27. L333. What is MS222?

28. L333-6. Was the dissection done at the catching site, or later in the lab? In the latter case, what was the time between catching the larvae and freezing the intestines, and how were the killed larvae transported to the lab (e.g. on ice)?

29. L342-344. It might be more clear to also provide the total number of larvae that were involved in this transplantation experiment (n=160).

30. L355. How long would the transition from larvae to adults normally take? I assume they had not yet metamorphosed yet, but it would be nice to give that information here to get a sense of time scale.

31. L351. The current wording suggests that bigger larvae were able to escape; it might be best to leave out "the smallest".

32. L367. Typo: "broad"

33. L378. With "sequences were joined", do the authors mean that the paired-end reads were combined?

34. L379. Should “and open-reference” read “an open-reference”?
35. L380. Please provide the percentage sequence identity that was used as a cut-off here, so that readers don’t need to go to the supplemental to find this important piece of information.
36. L390. 148 samples does not match layout provided in Figure 1 (160 samples). The reason is stated in L209, but it might be worth repeating here too.
37. L409. Comma between tool and PICRUSt can be deleted.
38. L416. Could the text here refer to the Supplemental materials, as done in L428/9, to clarify?
39. Figure 1. To avoid confusion with the transplantation schematic, it might be more clear to call the different sites e.g., Pond A, Pond B, and Stream A and Stream B, respectively. E.g in Figure 1A the left Pond could be called Pond A, and the right pond: Pond B.
40. Figure 2.
- The descriptions of A) and B) appear to have been switched. A) shows the alpha diversity, not the PCoA plot.
 - In C), could the authors please include the number of Core50 gut OTUs that were found? Suggested wording: “profiles of Core50 gut bacterial OTUs grouped (n=x) at genus level”.
 - In Figure 2C, it would make more sense to switch streams (currently on the left) and ponds (right) as to be consistent with 2A, other figures (e.g. S2), and the text.
41. Figure 4.
- Capitalization of PERMANOVA differs between figure panels.
 - Would it be helpful if the red (pond to pond) symbols were closed to, as to match the closed blue stream to stream ones?
 - The description of what is shown in 4A and 4C did not quite match up between text and panels. For example, Fig. 4A not only shows “larvae derived from streams and transplanted to ponds”, but also stream-stream and pond-pond larvae.
 - I am wondering if it would make more sense to combine Figure 4A and 4C (as done with the predicted functions in E). It took me a while to realize that only 3 of the 4 groups were displayed in each panel. Did the authors consider combining them, or was there particular reason to split them up?
42. Figure 6.
- The green color in panels A and C did not appear to match that of the legend. Even better, the authors could use the same colors as in Figure 4, but add e.g. “Skin” to the PCoA plot (and “Gut” to Figure 4). The latter alternative would make the skin vs gut shift similarities more obvious.
43. Supplemental Methods:
- In DNA Extraction: how were the skin swabs extracted? The protocol here only mentions the intestinal samples.
 - In DNA Extraction: “DNA extracts were stored at 20C” – should this be -20?
 - What is “HF buffer”? Manufacturer?
 - What is “Phusion Tag”? Should this be “Taq”?
 - Sequence Processing: Use “16S rRNA gene”
 - Please include a citation or website for the PICRUSt tool (as done in L397 of main text)

g. In Core Calculations, bottom of page 9. Do the authors mean “50% of individuals from either ponds or streams” (I think so)? The current wording was not clear. How many OTUs were in the Core50 from the field experiment?

Manuscript NCOMMS-16-18578 Response to Reviewer Comments

General Response: Thank you very much for the positive and constructive feedback on our manuscript. Your comments have greatly improved the manuscript. Below you will find our point by point responses to each comment. Please note that the line number reference refers to those in the clean (non-track edited) version of the manuscript. We also have submitted a track-change version of the revision.

Reviewers' comments:

Reviewer #1 (Remarks to the Author):

The original study by Bletz et al., "Gut microbiota of amphibian larvae shift differentially in community structure but converge on habitat-specific predicted functions", is an excellent contribution to microbial ecology with well-designed field experiments, and state-of-the-art analyses. The authors are able to demonstrate with a reciprocal transplant experiment that gut microbiota either shifts to match the microbial functions of destination habitat and diet, and this shift can occur either by a shift to match destination community composition, or a shift in community composition that doesn't match the destination, but still shifts to match the functional profile of the destination. Thus, functional profile is not tied to specific microbial composition, and the authors suggest that historical contingency and colonization resistance can explain incomplete community composition shifts. They also contrast gut community shifts with skin community shifts. The abundance of data generated by this study is clearly and well presented.

Response: Thanks for your positive feedback.

Specific comments and room for improvements:

Abstract line 39 delete "fully" and is there space to mention that skin communities were also measured for comparison with gut communities?

Response: "Fully" has been deleted. We see the point that the analysis of skin microbiota as a reference system should be mentioned in the abstract. We have adjusted the abstract to incorporate the skin microbiota and still meet the required word length of 150 words. (lines 29-38)

Line 51 change to "Host factors such as stomach pH, mucuins,..."

Response: This change has been made. (line 46)

Line 82 There's no such thing as 'rDNA', don't use this term

Response: This has been changed to rRNA. (line 70)

85 change to "understanding exists of the external environment's influence of these communities and of how..."

Response: This change has been made. (line 74)

87 change to "...organisms. Currently, we are lacking ..." this sentence should be re-worded, we will probably never have a "complete" understanding.

Response: This change has been made. (line 77)

10 delete “as a valuable tool”

Response: This whole sentence has now been deleted.

112 delete “This community...” and reword “Amphibian microbiota are well studied, and known for defensive ...”

Response: We have adjusted the wording to say “Amphibian skin microbiota is...” to make it more clear. (line 99)

114 change to “considered to be more...”

Response: This change has been made. (lines 100-101)

116 delete “abiotic and biotic” change to “which have additional nutritional (i.e. diet) influences.” Stress and the immune system are biotic influences on both skin and gut body sites.

Response: This change has been made. (line 103),

199 change to “larvae, gut...”

Response: The structure of this sentence has now changed. (lines 105-106)

123 and everywhere else delete “complete”. I don’t think you show complete matching, but rather not significantly different?

Response: We had used this wording of “complete” and “full” to give emphasis to the observed patterns. However, we understand your point, and have removed these throughout the manuscript.

162 Here summarize the differences in function that are described in the discussion.

Response: We have added a few sentences to describe the functional differences between pond and stream larvae. We have kept this short to avoid simply listing all the features presented in Table S1, and to avoid over interpretation of our PICRUSt results. (lines 155-161)

Section starting line 179 – here it is not clear why there would be any changes in pond-to-pond transplants or in stream-stream transplants. Two kinds of analyses would help explain this, or provide interesting additional results: 1. Describe in the text the results from fig. 4 showing little difference in stream to “native stream” vs. stream to “non-native” stream. Same for ponds. 2. What were the changes from field to semi-captive experimental conditions? Did diversity decrease after putting larvae in enclosures? Do enclosures/different food access explain these small pond-pond and stream-stream differences? This could be mentioned in text with a fuller analysis in the supplement.

Response: Perhaps there has been a misunderstanding of the LEfSe analyses that we performed with the transplant experiment data. The idea of the factorial transplant experimental design was to have larvae being switched from one habitat type into the other (i.e. from stream to ponds and vice versa) and as a control, larvae that were moved between the same habitat type (i.e. from streams to streams and ponds to ponds). What we have tested with the LEfSe analyses here is based on the hypothesis that gut bacterial communities of habitat-switched individuals will be different from their **origin** habitat controls. This is why we would predict a shift in the OTUs present when larvae are switched into to the “non-native” habitats. If true, there would be differentially abundant OTUs between these groups. What we have compared here was two-fold: (1) pond-to-stream was compared to pond-to-pond, and (2) stream-to-pond was compared to stream-to-stream. We have expanded the introductory sentence to this paragraph in attempt to make the nature and purpose of these LEfSe analyses more clear. We hope this additional information clears up the purpose of this analysis. (lines 190-193)

The potential effect of enclosures is indeed an important point. A small number (n=10) of free-swimming larvae were sampled on the same day as the experimental larvae collection. Accordingly, we have carried out PCoA visualization, PERMANOVA analyses, and alpha diversity (Chao1 and PD) comparisons to illustrate that the enclosures did not have or only had minimal effects on the gut and skin microbiota. These results are now included as a supplemental figure (Supplementary Figure 3), and a brief summary of these results is now included in the main text results section about the transplant taxonomic community data (lines 182-189).

231 change “naturally” to “natural”

Response: This change has been made. (line 262)

249 change to “intake and digestion of different...” Microbiota are not changing because the host “needs” them to.

Response: This change has been made. (line 289)

252 delete “required” and change to “likely induced bacteria with different suites of enzymatic activities, and promoted host digestion.”

Response: This change has been made. (line 289)

255 Some of these metabolism differences should be in the results section also.

Response: Agreed. See response to similar comment in the result section above.

280, also 288 and “full” in 291 – Why would you expect a complete shift?

Response: Based on the comment above we have deleted occurrences of “full” and “complete” through the manuscript. The reasoning behind the expectation of a shift is provided in the Discussion (lines 291-301), where we have stated: “Given the importance of abiotic and biotic environmental conditions in shaping gut microbiota from previous studies comparing different hosts, it was predicted that individuals transferred into the non-native habitat (i.e. habitat-switched individuals) would shift and exhibit gut bacterial communities similar to individuals naturally found there.”

Define complete. Do you mean that a shift occurred so that the transplant community was not significantly different from the control community?

Response: Based on the comment above we have deleted occurrences of “full” and “complete” throughout the manuscript, and therefore we have not included a definition of this in the main text.

281 change to “to that of larvae”

Response: This change has been made. (line 308)

295 delete “flora” since you are not studying plants. “biota” would be appropriate.

Response: This has been changed to “taxa”. (line 329)

302 How does this compare with findings of Kueng et al. 2014 Stability of microbiota facilitated by host immune regulation: informing probiotic strategies to manage amphibian disease. PlosOne, where adult amphibians showed skin colonization resistance to disturbance?

Response: Thanks for drawing our attention to this study. We have incorporated this into the discussion, mentioning the concept of host immune regulating facilitating resistance to disturbance. (line 340)

315. Why only shotgun? Wouldn't targeted 16S rRNA transcriptomics also work to get at dormancy?

Response: We have adjusted the wording to include 16S rRNA-targeted transcriptomics. (lines 357-357)

I suggest a table or revised fig 5 that includes % OTUs that significantly change.

Response: Thanks, for this good suggestion. We have now added in the number of differential LEfSe taxonomic features to parallel the differential metagenomic features. The total number of OTUs/features going into these analyses are provided in the legend. (See Figure 5)

367 change board to "broad"

Response: This change has been made. (line 437)

Figs 2 and 4 – the letters on figures do not match the legend text.

Response: Thanks for noticing this. We have fixed the labels to correspond correctly. (See figures legends for Fig 1 and 4 now; lines 703,722).

Fig 4 legend. 644 Do you mean "identical" or not sig. different?

Response: We have added a note to define "identical" as "not significantly different" within the figure legend. (line 737)

When discussing, I would expect that salamanders have a core set of bacterial OTUs that stay constant among all conditions and define the species microbiome. Species identity is thought to be a stronger predictor or driver of microbiota than environmental conditions, at least for the skin. Can you comment on this? What percent of OTUs differed between habitats?

Response: Yes, this is an important point. We have added two sentences in the results to highlight the overlap in the gut community core50 for pond and stream larvae. (lines 145-148). We have also added a couple sentences to the discussion regarding the core gut microbiota. (lines 272-277)

Fig. 5 I would like to see, in addition to features, the % difference in terms of proportion of OTUs based on the LEfSe analyses.

Response: Yes, this would be a good addition, thanks. We have added in the number of differential LEfSe-detected taxa to parallel the differential metagenomic features in Figure 5. The total number of OTUs/features going into these analyses are now provided in the legend. (See Figure 5)

An additional part to this figure, or a supplement could indicate the same things in the gut and skin for field pond to enclosure pond, and field stream to enclosure stream. There are some interesting changes occurring, just by putting the larvae into the enclosures. Are there enclosure specific OTUs?

Response: As mentioned above, the potential effect of enclosures is indeed an important point. Based on our analyses, the impact of the enclosures is minimal, if present at all (see above for more details) and these data are now provided in Supplementary Figure 3; therefore, we do not see the LEfSe analyses of this data as a needed addition. We hope you will agree.

Reviewer #2 (Remarks to the Author):

In this study, Bletz et al. analyze the effects of the ecological niche on the structure, composition and function of gut microbiomes in an interesting animal (amphibian) model – Salamandra

salamandra. The authors designed an ecological transfer experiment, where pond-adapted salamanders were transferred to a stream environment, and vice versa. This interesting experimental design allowed the authors to measure the impact of both abiotic and biotic parameters of the ecological niche on the gut microbiomes.

Overall, I have found this paper a stimulating read, with an interesting model (the two salamander populations have recently undergone ecological differentiation), a well-conceived experimental setup and stimulating biological conclusions. The paper will be of interest to the community of microbial ecologists.

Response: Thanks for your positive feedback on our study.

However, I think that many parts of the manuscript need to be improved or clarified, and some of the statements need to be toned down.

-- Main concerns

1. The authors are not measuring the functional profiles of these microbiomes with metagenomics sequencing. Instead, they use 16S and an in-silico program to predict functions based on the taxonomic profiles of these communities. To me this is fine as long as the authors admit and discuss the limits of the approach. Instead, they attempt to convince the reader that it is better than doing metagenomics to characterize functional profiles. Even though metagenomics can “overlook functions from less abundant members” (line 76), it is also the case with 16S. 16S amplicon sequencing also depends on pseudo-universal primers that actually miss a large component of the microbial biodiversity, which has been shown in many studies. Metagenomics is much less affected by this issue.

Furthermore, it is stated that host DNA could negatively impact the sequencing of microbial communities and subsequent analyses (line 76-78). This is very controversial, and there is no general rule for this. The reference cited at this stage to support this claim is not convincing in this context.

Still around this issue, the PICRUSt program has its own limits too, especially when applied to non-human hosts. As it predicts functions, there is uncertainty on these predictions, which should also be acknowledged. Langille et al. showed in their original paper that PICRUSt is very accurate at predicting functions for human microbiomes, but that it is much less accurate to do so in other non-human mammals. What stands for non-human mammals is also very likely to stand for amphibians, as the reference genomes used by PICRUSt are biased towards human-associated microbes.

Overall, I don't think the authors should be criticized for not having performed metagenomics, especially because their conclusions on functional profiles make sense regarding the original hypotheses. But I think the introduction needs to be rephrased and that conclusions drawn from PICRUSt experiments should be discussed to incorporate these remarks.

Response: We completely understand. We have altered the introductory material about PICRUSt in concordance with these remarks. We have removed the material saying that metagenomics can overlook low abundance functions and the text describing the host DNA issues (lines 65-72). Additionally, we have added phrases to the Discussion stating that PICRUSt has limitations (lines 305-306 and 321-324). We further expand on this in the main text methods where we have given the NSTI value, which is a measurement of how well the taxa in our samples are represented in the genome databases (lines 481-487). This information was previously in the supplemental material. We additionally point out that we are using PICRUSt solely in a comparative context in our system not with the goal of describing functions which we believe limits some of the typical concerns with PICRUSt as any bias if present is equally associated with the comparative groups of our study organism.

2. To what extent the gut microbiome of salamander larvae reflects the microbiome of its aquatic environment? One potential issue here is whether the authors are observing profound changes in the microbiome because new bacteria are colonizing per se the gut after the transfer to a pond or a stream, or whether the bacteria are transiently passing through and are actually living in the water. The skin microbiome experiment can control for this to some extent, but the skin in amphibian is also known to host a specific microbial community that is not entirely influenced by the surrounding water. The authors should discuss this.

Response: Yes, we understand your point perfectly. We have added a supplemental figure (Figure S5) which includes taxonomic summary plots of environmental samples from both stream and pond water collected during the field survey. This shows that the gut microbiota of salamander larvae do not simply reflect the microbiota of the surrounding aquatic environment. We have added a brief sentence to the results highlighting this result (lines 126-127). Of course we cannot fully rule out that some of the bacterial sequences are transient bacteria, but the different community compositions of the gut and aquatic habitat can demonstrate that we most likely are not observing patterns of bacteria that are transiently passing through.

3. The authors name their experiment a “reciprocal transfer” at line 107, which I think is good. But then in the rest of the manuscript, the terms “transplant”, “transplanted” and “transplantation” are used. To me this is confusion with the gut microbiome literature, in which the word “transplant” is used at length to talk about microbiome transplant (such as Fecal Microbiota Transplantation (FMT)), which is also mentioned by the author at line 53. When reading the manuscript, it took me time to realize that the author are transferring or transplanting individuals into different environments, and not transplanting bacterial communities themselves. I think the word “transfer” should be preferred in this manuscript to avoid confusion, especially in the context of this literature.

Response: Thanks for this important advice! We have adjusted all occurrences of “transplant or transplantation” to be “transfer” to avoid any misunderstanding of terminology used in the present literature.

4. There are many issues with the figures and their legends:

- First, Figure 1 is cited in the text after Figure 2 and 3. As Figure 1 presents the ecological transfer experiment, it should be placed as figure 3.

Response: Thanks. We have fixed the order of figures accordingly to this comment.

- Line 206: Figure 7 is cited, but this is Figure 6.

Response: Thanks for noticing this. We have fixed the reference to correspond correctly. (line 219)

- In Figure 1 legend: please explain what P.KoE, P.KoV etc means. Even if these acronyms are explained in the Methods, it would help the reader to have this info right below the figure to more easily understand the experiment.

Response: Thanks for the advice. An explanation of these labels has been added to the legend. (line 721)

- Figure 2B: In this PCoA plot, the two axes are “PCO1”. I guess the y axis represents the second component of the PCoA?

Response: Yes, you are correct. This has been fixed. (See Figure 1)

- Figure 2 legend: A) and B) do not point to the right plots. A) should be the alpha diversity and B) should be the PCoA.

Response: Thanks for noticing this. We have fixed the labels to correspond correctly. (starting line 703 (now Fig 1))

- Line 625: analyses -> analysis

Response: This change has been made. (line 697)

5. This paper is also interesting in the context of biological adaptations promoted by gut microbiome plasticity. Few studies have designed experiments to measure the benefit that gut microbiome rearrangements provide to the host when the host faces new environmental conditions. In this context, the measurement of growth rate following the ecological transfers is interesting. The higher growth rate observed for the stream-to-pond individuals is intriguing, because it corresponds to the higher growth rate observed among individuals living in pond environments. However, it is difficult to conclude whether the microbiome rearrangements following the stream-to-pond transfers is causing this higher growth rate, or whether it is simply the effect of the higher temperature in pond waters. An ideal solution would be to obtain germ-free larvae, and compare the growth rate of larvae living in hot (pond) vs. cold (stream) temperatures, and measure the benefit of the microbiome with respect to this baseline difference. Maybe the authors would be interested in discussing these topics in their Discussion section? (Especially because at the moment the discussion unnecessarily reiterates some of the main conclusions stated in the Results).

Importantly, several papers that are relevant to this study are not cited and the results should be discussed in the context of these papers, to feed the discussion regarding the impact of environmental change on the gut microbiome:

- Alberdi, A. et al. (2016) Trends in Ecology and Evolution. <http://dx.doi.org/10.1016/j.tree.2016.06.008>
- Sommer, F. et al. (2016) Cell Rep. 14, 1655–1661
- Chevalier, C. et al. (2015) Cell 163, 1360–1374

Response: This is definitely a very interesting point! Thank you for bringing up this point and providing these important references; the very recently published Alberdi et al. paper, especially, provides a very interesting background context for our study. Specifically related to the results of growth rate and also the stomach content analysis, we have mainly included these pieces in order to demonstrate that larvae have really eaten and preyed on habitat-specific food items in the streams and ponds when being transferred. However, we see the relevance of discussing these results in a bit more detail.

We agree with integrating this idea of gut microbiota promoting adaptation/acclimation into the manuscript, and have added text to the manuscript to accomplish this. First, we have briefly highlighted/introduced the idea that gut microbiota may be an important factor in processes of ecological adaptation in the introduction (lines 43-44) and discussion (lines 258-260, line 265-66). We further expand on this concept in relation to our data in the discussion (paragraph starting at line 367). Right now, in the discussion we first propose that the existence of functional shifts (albeit through different taxonomic means) could be seen as “metagenomic plasticity” (as defined by Alberdi et al.). We then connect this with our result that there is no reduction in growth rate seen in comparison to destination habitat controls, which are not compromised by the temperature differences between habitats. With this approach we feel we are being a more conservative. Finally, we have reduced the re-presentation of results at the beginning of the discussion.

6. The authors find that pond-to-stream larvae converge in functions but not in taxonomy with stream larvae. This is interesting, but the authors do not explain how this happens. They propose different hypotheses in the Discussion (line 276-280), but it is not obvious to me why

they cannot test them. What are the different taxa that are functionally redundant? Are they phylogenetically distantly related?

Response: We too were interested in this question as the conclusions of our experiment unfolded; however, identifying specific OTUs to address such questions is a complex task. While conceptually simple (is Function1 in S>S associated with OTUA and Function 1 in P>S associated with this same "OTUA" or a different one, perhaps OTUB?), the nature of our data is always embedded in a community context, which, in our view, makes definitively identifying OTUs that are functionally redundant quite complex. We have explored such an approach by using PICRUSt. Within PICRUSt there is a script called "metagenome_contributions.py" which associates K#s (i.e. the functional units) with the OTUs in your samples; however, it is not a one-to-one relationship; each K# can be associated with multiple OTUs (2-60+ based on trial analyses with our data) and each of the connections then has a proportional contribution value. We are not only dealing with presence/absence of an OTU providing functions but also the relative proportion of a function that is provided by the OTUs, based on their relative abundance in the samples. Furthermore, there are almost 5000 K #s. Given this complexity, we cannot see a feasible way to get at the OTUs driving this functional matching or which OTUs may be functionally redundant. We could select focal functions to look at, however, this leads to the question of how do we decide the functional categories or specific K#s to investigate? Will these be representative of the full dataset? As a trial analysis we did explore all K#s associated with the functional category of "Lipid Metabolism". This group alone contains 150 K#s in our dataset. There are both some OTUs that are only providing a given function within one group while other OTUS contribute to a function in both experimental groups but differ in their proportional contribution, which shows how difficult it seems to address these hypotheses in detail. We would be happy to prepare and provide a document containing our trial analyses of these data upon request.

It may be likely that both of the hypotheses we present are at play. Some portion of the function(s) are obtained by OTUs that are "picked up" by the P>S group (that are also in S>S controls) while another portion of the function(s) is provided by the "different" taxa between the groups, and therefore it may be only when both of these are put together that we get functional matching (i.e. the "community" context of it matters). However, given the complexity of these data and doing this for the full functional dataset, we are wary to make definitive conclusions about these hypotheses, and would rather leave the discussion as is (lines 319-321), presenting the possible hypotheses, and postpone elucidating such hypotheses to a stage in which true metagenomic data become available. We hope that you understand our opinion on this, and if there are tools available that we are not aware of, we will be happy to receive input on this.

Importantly, is the conclusion that taxonomic convergence is not found dependent on the similarity threshold that is used to define bacterial entities (97%)? Can convergence be seen at higher taxonomic levels? (lower similarity cutoffs, say 95, or 93%?). If yes, it would then indicate that taxonomic signatures of pond vs. stream are simply determined at different taxonomic levels along the bacterial tree, and that searching for convergence at 97% for stream larvae is bound to produce divergent signatures (while still producing convergent functions).

Response: To address this, we completed the PCoA visualization and PERMANOVA analysis on beta diversity matrices originating from OTU tables at all taxonomic levels (Genus, Family, Order, Class and Phylum) (now provided as Supplemental Figure 6) for the transplant experiment data, and the divergent signatures of the taxonomic community structure are maintained across the levels. This result suggests that our results are independent of the similarity threshold used. We have added a sentence in the results to incorporate this finding. (lines 178-179)

-- Minor concerns

Line 72: In addition, -> In addition

Response: This has been fixed. (Line 64)

Line 88: “complete” lack of understanding is too strong.

Response: This has been changed to “in depth” to reduce the strength of this statement (line 77)

Line 147: As this is the first time you mention “LEfSe LDA” in the manuscript, please spell it out so that the reader can immediately understand what you did.

Response: The full name is now included. (line 136)

Line 156-162: this paragraph is a little weak with no clear conclusion. Please expand with more details and impact.

Response: We have added some additional sentences describing a few of the differential features that were found to strengthen this paragraph. (Paragraph begins on line 149)

Line 279: I don’t understand what this sentence means.

Response: The purpose of this sentence is to point out that the skin and gut communities did not follow the same patterns and to transition into the next sentence which summarizes what was seen in the skin communities of transplanted communities. We have restructured this sentence now in effort to make this clearer. It now reads: “The response of the skin microbial communities of the same transplanted individuals further highlights the unique nature of the differential response seen in the gut microbiota.” (lines 306-308). We hope that we have expressed our statement more clearly now.

Reference 10: animalcules -> animalicules

Response: We have fixed this citation.

Signed: Mathieu Groussin

Reviewer #3 (Remarks to the Author):

To address the influence of environmental factors (both diet as well as abiotic factors) in the development of host-associated microbial communities, the authors analyze gut and skin communities from salamander larvae collected at two different aquatic habitats: ponds or streams, and those from larvae transplanted to the other habitat. In addition to a 16S rRNA analysis, they used PICRUSt to infer functionalities of the microbial communities.

There has been very few papers on the salamander microbiome, in particular that of the gut, so this paper is novel and will provide much needed data for this understudied group.

Most graphs are well made, and text and experimental set up are for the most part clear. I have mainly minor comments.

Response: Thanks for your positive remarks.

General comments

1. The Introduction felt very long – the first 2 paragraphs could maybe be summarized in 2-3 sentences about the essential role of the gut microbiota and possible factors that influence its composition.

Response: We have reduced the text of the 2nd paragraph in the introduction in an effort to address this comment. (lines 45-52)

2. How do the communities found in/on salamander gut and skin compare to previously published data in salamanders or frogs? The authors could first point out that not many studies have been performed on this animal group, and that this study fills that gap, even without performing the transplant analysis. The authors could then compare their skin microbiota data to that published in e.g. doi: 10.3354/dao02004 and 10.1038/ismej.2007.110 (both not by this reviewer). Do multiple studies find the same bacterial groups?

Response: We have added a supplemental figure to display the overall composition of stream and pond larvae gut and skin communities at the phylum level. In the Discussion, we use this to briefly compare to other known studies for amphibian larva with focus on the gut community because this is the central focus in our manuscript. The phylum groups found in salamander guts fit well into what has been described for other amphibian larvae (Vences et al 2016, Kohl et al 2013, Chang et al 2016). Additionally, in the results we have added a sentence briefly mentioning the taxonomic composition of the skin microbiota (line 218-221), but in order not to lose focus of our study we have avoided a detailed discussion of the skin microbiota. However, for those that are interested in a detailed characterization and comparison, we provide a reference to a detailed study of the skin microbiota of fire salamander larvae and adults by our group, which is in press in *Microbial Ecology* (line 222).

3. How many of the 16S rRNA gene OTUs were new, i.e. not seen before in previous studies? Given the sparse data on salamander microbiota, I would expect that this study has found many novel taxa. Could the authors do e.g. a BLAST search and report the percent of OTUs that are <97% identical to known sequences? Alternatively, they could use the percent of OTUs that did not have a >97% match in the reference database during OTU binning. This would be a nice datapoint to add to this study.

Response: We have added the percentage of *de novo* OTUs present within the full gut community and the core-50 gut community to the results section (lines 120-122). Here we focus on the gut microbiota as this is the main focus. As mentioned above, there is a recent publication in press in *Microbial Ecology* which characterizes fire salamander skin microbiota, and contains more detailed analyses, such as this, for skin microbiota.

4. PICRUSt. Maybe the authors could add some wording to express caution about the use of PICRUSt in this particular setting. It performs well in case of well-characterized and genome-sequenced microbial species, but how well would it perform with microbial species from a less well-studied sample type such as salamander gut? PICRUSt relies heavily on the presence of sequenced genomes, and I am not sure if it would perform very well in a situation where not many microbial members of a community have been sequenced.

Response: We understand these concerns and note that major comment 1 of reviewer 2 also refers to this aspect. We summarize our response here again: We now give the NSTI values, which is a measurement of how well the taxa in our samples are represented in the genome databases, in the main text methods (lines 481-487). This information was previously in the supplemental material. We additionally point out in the methods that we are using PICRUSt solely in a comparative context not with the goal of describing function which we believe limits some of the typical concerns with PICRUSt as any bias, if present, is equally associated with our comparative groups. In addition, we highlight in the Discussion that true metagenomics or transcriptomics are needed to verify our findings (lines 305-306 and 321-324).

5. Where all larvae from the same salamander species? Could the larvae belong to multiple

species or was species determination easy to do? Could the species (mentioned in the Introduction) also be added to the Methods?

Response: Yes, all salamander larvae are from the same species – the fire salamander *S. salamandra* – and are unambiguously identifiable from co-occurring newt larvae if present in the same habitat. We have added both the species name and a sentence about this to the Methods of the main text (lines 380, 383-384). In addition, we have added a section to the supplemental methods describing the species and study system in a bit more detail. This can act as a supplement to the brief description currently in the introduction.

6. Could the higher "resistance" to change of the pond-to-stream group be explained by the lower species richness/diversity of the pond communities as shown in Figure 2? This might be an obvious, yet overlooked explanation (maybe similar to extinction of microbes found in subsequent generations when given a low-fiber, and then a high-fiber diet – once gone, they cannot revert back, Sonnenburg, doi:10.1038/nature16504, not by this reviewer). Maybe the authors could add this thought to the end of the Discussion.

Response: Thanks for bringing our attention to this reference. We agree this may be a possible explanation and have incorporated this now into the discussion (lines 333-337).

7. No accession number has yet been provided, so this review could not evaluate these.

Response: We have now included the SRA accession numbers for these data. (lines 509-510).

Specific comments

8. L47. I'm not a big fan of the "forgotten organ" term – microbiologists have long recognized that gut microbiota plays an important role in digestion, and it is not an organ either.

Response: The phrasing here has been reworded to say "...play a fundamental role..." (lines 40-41).

9. L58. "select the composition of the microbial community" or "influence the composition..." would be better here.

Response: The wording has been changed accordingly to "...strongly select the composition of the microbial...". (line 49).

10. L60. Although I agree that temperature could be considered an abiotic environmental condition, it could also be a host-associated factor (such as those listed in L51), at least in e.g. mammals and birds. This text felt very long and split up into not-very-clearly defined groups.

Response: While we understand your point, given that amphibians are ectothermic and cannot regulate themselves their temperature, we have chosen to keep the temperature reference associated with abiotic environmental conditions. We have added a phrase to clarify this distinction (line 52). Additionally, we have worked to condense this section of the introduction to make it more streamlined.

11. L66. Can you really say that a host "acquires" a new habitat?

Response: We have changed the wording to "entry into" a habitat. We hope that this term is seen as more appropriate for this sentence. (line 57)

12. L71 and other places: Please use "16S", with a capital S (Svedberg).

Response: This has been corrected throughout the manuscript.

13. L72. "In addition to the analysis of the structural response, it is perhaps even more important to study whether such..." would be better.

Response: This has been change accordingly. (lines 63-64)

14. L82. This should read “16S rRNA gene”.

Response: This change has been made. (line 70)

15. L102. Please use “16S rRNA gene amplicon”

Response: This change has been made throughout the manuscript.

16. L109. “We utilized...” sounded similar to what was already stated in L104-107.

Response: Yes, we agree, this is somewhat repetitive. We have removed this sentence.

17. L112. To what does “this community” refer? To the communities of salamanders or to skin communities? Please clarify.

Response: We have rephrased this sentence to define what “this community” was. It now reads “Amphibian skin microbiota is...” (line 99)

18. L118. The last part of the Introduction, starting with “We found...” could maybe be deleted here and integrated into the Discussion.

Response: This brief results summary is as requirement on the Nature Communication check list we have received. Therefore, we need to maintain this section in the text.

19. L121. Why was this unexpected? The previous text in the Introduction suggested that it would change. The shift itself was not unexpected, because communities were expected to shift, but maybe the authors meant that the direction/nature of the shift was unexpected?

Response: Yes, we are meaning the directionality of shift here. This part of the paragraph has been adjusted to clarify its meaning (lines 107-111).

20. L129. Please use “16S rRNA gene amplicon”.

Response: This change has been made.

21. L135-138. Since Figure 2B only shows one of these three distance metrics, this should probably be worded differently. It currently sounds as if 2B will show all three analyses.

Response: The figure reference has been moved to directly follow the Unweight Unifrac statistics, which is the metric presented in the figure. (line 125)

22. L172. The current placement of the Figure 1 referral suggests that Figure 1 shows the responses. Suggestion to change text to: “reciprocal transplant experiment (Fig. 1) showed distinct responses”.

Response: This has been changed accordingly. (line 171)

23. L204. Would “shifted completely to match that of the destination habitat control individuals” be more clear?

Response: Yes, indeed. This has been changed accordingly. (line 217)

24. L206. Should this be Figure 6?

Response: Yes, thanks. This has been fixed. (line 219)

25. L327. Please add more information about the species of salamander that was used in this study. Is this an endangered species? Did the authors need any permits or other permission to catch salamanders? Also, where these larvae collected around the same time as the larvae for the transplant experiment (i.e., May 2015)?

Response: The Fire Salamander is a protected species under German law and listed by the IUCN under the category ‘Least Concern’. Given that fire salamanders are quite abundant in the Kottenforst, we received permission by the Untere Naturschutzbehörde der Stadt Bonn to sample larvae for our study. We have mentioned this permission in the acknowledgements, but we have now included a statement in the material and methods section (lines 391-394) to show that appropriate permission was received. Additional information on this species and study system has been added to the supplemental methods in effort to keep the main text methods concise. Dates are provided in the Field Survey and experimental methods showing that the Field Survey individuals were caught in late April of 2015 and the transfer experiment was performed in May 2015. (lines 396, 423).

26. L329. Please add the location where the Kottenforst is situated (e.g. near the City of Bonn, Germany).

Response: “(near the City of Bonn, Germany)” has been added as suggested. (line 377).

27. L333. What is MS222?

Response: The full name, Tricaine Methanesulfonate, has been added to the text to clarify this. (Line 386)

28. L333-6. Was the dissection done at the catching site, or later in the lab? In the latter case, what was the time between catching the larvae and freezing the intestines, and how were the killed larvae transported to the lab (e.g. on ice)?

Response: Larvae euthanization and dissection was performed in a field station/laboratory. We added this information to the Methods of the main text. It now reads: “Altogether, 104 (60 stream and 42 pond) larvae were collected, transported live to the field station/laboratory, and immediately (within 5 hours of collection euthanized with Tricaine Methanesulfonate (MS222). Subsequently, larvae were dissected and the stomach and intestine portion of the digestive tract were separated. Stomachs were preserved in 80% ethanol for visual food spectrum analysis, and intestines were frozen at -80°C immediately after dissection and stored until DNA extraction for gut bacterial community analysis.” (lines 398-404).

29. L342-344. It might be more clear to also provide the total number of larvae that were involved in this transplantation experiment (n=160).

Response: The total number has been added to the main text as suggested. (line 411).

30. L355. How long would the transition from larvae to adults normally take? I assume they had not yet metamorphosed yet, but it would be nice to give that information here to get a sense of time scale.

Response: Time to metamorphosis strongly depends on abiotic (e.g. temperature) and biotic (e.g. food availability) factors. On average, larvae need two to three months until metamorphosis in the stream habitat and less time in the pond habitat. For our experimental setup, we used larvae at an early developmental stage so that they did not enter metamorphosis during the 2-week transplant experiment. We have clarified this now by an additional sentence. (lines 423-435).

31. L351. The current wording suggests that bigger larvae were able to escape; it might be best to leave out “the smallest”.

Response: “smallest” has been deleted as suggested.

32. L367. Typo: “broad”

Response: This typo has been fixed (line 437).

33. L378. With “sequences were joined”, do the authors mean that the paired-end reads were combined?

Response: Yes, you are correct. We have adjusted the wording to say “Briefly, paired-end sequences were merged” to make this clearer. (line 434)

34. L379. Should “and open-reference” read “an open-reference”?

Response: Yes, correct. This has been fixed. (line 448)

35. L380. Please provide the percentage sequence identity that was used as a cut-off here, so that readers don't need to go to the supplemental to find this important piece of information.

Response: The percentage threshold was 97% and this has now been added into the main text as suggested. (line 449)

36. L390. 148 samples does not match layout provided in Figure 1 (160 samples). The reason is stated in L209, but it might be worth repeating here too.

Response: We added a parenthetical phrase here saying “... 148 samples (total number of individuals that survived the experiment)” to reiterate this here. (lines 459)

37. L409. Comma between tool and PICRUSt can be deleted.

Response: This comma has been removed. (line 466)

38. L416. Could the text here refer to the Supplemental materials, as done in L428/9, to clarify?

Response: Yes, a reference to supplemental materials has been added accordingly (line 479)

39. Figure 1. To avoid confusion with the transplantation schematic, it might be more clear to call the different sites e.g., Pond A, Pond B, and Stream A and Stream B, respectively. E.g in Figure 1A the left Pond could be called Pond A, and the right pond: Pond B.

Response: Thanks for this suggestion! We have updated the figure accordingly.

40. Figure 2.

a. The descriptions of A) and B) appear to have been switched. A) shows the alpha diversity, not the PCoA plot.

Response: Thanks for catching this. We have fixed the A) and B) reference to be correct. (starting line 703, now Figure 1)

b. In C), could the authors please include the number of Core50 gut OTUs that were found? Suggested wording: “profiles of Core50 gut bacterial OTUs grouped (n=x) at genus level”.

c. In Figure 2C, it would make more sense to switch streams (currently on the left) and ponds (right) as to be consistent with 2A, other figures (e.g. S2), and the text.

Response: Thanks for catching this. We have fixed the A) and B) reference to be correct. We also added in the total number of OTUs present in the Core50 into the legend as suggested. Lastly, we have switched the taxa plots of the ponds and streams to be consistent with 2A as suggested. (line 703)

41. Figure 4.

a. Capitalization of PERMANOVA differs between figure panels.

b. Would it be helpful if the red (pond to pond) symbols were closed to, as to match the closed blue stream to stream ones?

c. The description of what is shown in 4A and 4C did not quite match up between text and panels. For example, Fig. 4A not only shows “larvae derived from streams and transplanted to ponds”, but also stream-stream and pond-pond larvae.

d. I am wondering if it would make more sense to combine Figure 4A and 4C (as done with the predicted functions in E). It took me a while to realize that only 3 of the 4 groups were displayed in each panel. Did the authors consider combining them, or was there particular reason to split them up?

Response: We have made the figure panels consistent with “PERMANOVA”. We have also followed the suggestion of making the red pond symbols solid so both habitat controls are solid, and consolidated everything for the gut bacterial community structure to be in one PCoA plot (i.e. 4 A and C were combined). We have adjusted the figured legends accordingly and work to make them more clear. Lastly for this main text figure we have also simplified the symbols to be one per treatment. This makes the interpretation easier when all data is presented on one PCoA.

42. Figure 6.

a. The green color in panels A and C did not appear to match that of the legend. Even better, the authors could use the same colors as in Figure 4, but add e.g. “Skin” to the PCoA plot (and “Gut” to Figure 4). The latter alternative would make the skin vs gut shift similarities more obvious.

Response: Indeed, it makes sense to use the same colors. We have adjusted the colors and presentation here to match that of the gut transfer experiment figure (now figure 4).

43. Supplemental Methods:

a. In DNA Extraction: how were the skin swabs extracted? The protocol here only mentions the intestinal samples.

Response: Skin swabs were extracted following the same method. We have now added this information to the supplemental methods.

b. In DNA Extraction: “DNA extracts were stored at 20C” – should this be -20?

Response: Thanks for catching this typo. We have fixed this to read “-20° C”

c. What is “HF buffer”? Manufacturer?

Response: This is the PCR Buffer Mix for the Phusion Taq used. We have added a description to the sup. methods along with the Manufacturer info. It now reads: HF Buffer (High Fidelity PCR Buffer, New England Biolabs (Ipswich, MA))

d. What is “Phusion Tag”? Should this be “Taq”?

Response: You are correct. We have changed this accordingly

e. Sequence Processing: Use “16S rRNA gene”

Response: This change has been made.

f. Please include a citation or website for the PICRUSt tool (as done in L397 of main text)

Response: We have added in the literature reference for PICRUSt as well as the website on the Galaxy platform within the sup. methods.

g. In Core Calculations, bottom of page 9. Do the authors mean “50% of individuals from either ponds or streams” (I think so)? The current wording was not clear. How many OTUs were in the Core50 from the field experiment?

Response: We have adjusted the wording of the Core calculation in effort to make this clearer. We have also added in the number of Core50 OTUs within each habitat type and in total for both the gut and skin communities.

Reviewers' Comments:

Reviewer #1 (Remarks to the Author):

The authors have made all the appropriate responses and edits according to reviewer comments. I have no more concerns.

Reviewer #2 (Remarks to the Author):

I still support this paper, which is scientifically sound and interesting in many ways.

The introduction is well written, and has been greatly improved in comparison with the first version.

Below are a few comments that I think could help improving the paper even further, but the editor and/or the authors can choose to ignore them.

Results:

L135: It appears that the gut microbiome of stream larvae are enriched in aerobic bacteria (Proteobacteria), while pond larvae are enriched in Firmicutes, which usually contain anaerobes. I think the authors should briefly comment on this. It strongly suggests that the environment deeply influences the type of bacteria that can colonize the gut of salamander larvae, much more than in mammals for instance, which live in aerobic conditions but which are mostly colonized by anaerobes.

L156 & 284-291: the authors should explain more explicitly the potential link between dietary items of each type of environment and the specific functional categories that are found associated with these environments. Are dietary items found in stream environments (amphipods, insects) more enriched in lipids than those found in ponds (ostracods, copepods, cladocerans), which would explain the differences of content in gene functional categories? Or is it better explained by abiotic factors specific to each environment?

Minor

L119: “denovo OTUs”. In the literature, the term denovo OTU is usually employed to describe an OTU that has been called with a denovo algorithm without the help of an external database. Here, the authors are referring to “OTUs with unknown taxonomic classification”. I think this sentence should be rephrased to avoid confusion.

L180: Chao1 diversity differed slightly between... => Chao1 diversity slightly differed between...

Mathieu Groussin

Reviewer #3 (Remarks to the Author):

The authors have addressed all reviewers' comments and I have no further comments or suggestions. I trust that all edits have been made as stated in the rebuttal letter and congratulate the authors with this new version of the manuscript.

Signed, Elisabeth Bik, Stanford University.

Response to Reviewers

We are grateful to the reviewers for their positive evaluation of our revised manuscript and for their final minor suggestions. We have addressed all but one of these and provide detailed point-by-point responses in the following.

REVIEWERS' COMMENTS:

Reviewer #1 (Remarks to the Author):

The authors have made all the appropriate responses and edits according to reviewer comments. I have no more concerns.

Reviewer #2 (Remarks to the Author):

I still support this paper, which is scientifically sound and interesting in many ways.

The introduction is well written, and has been greatly improved in comparison with the first version.

Below are a few comments that I think could help improving the paper even further, but the editor and/or the authors can choose to ignore them.

Results:

L135: It appears that the gut microbiome of stream larvae are enriched in aerobic bacteria (Proteobacteria), while pond larvae are enriched in Firmicutes, which usually contain anaerobes. I think the authors should briefly comment on this. It strongly suggests that the environment deeply influences the type of bacteria that can colonize the gut of salamander larvae, much more than in mammals for instance, which live in aerobic conditions but which are mostly colonized by anaerobes.

Response: This is indeed an interesting point. We have incorporated this idea into the discussion (Line 285-289).

L156 & 284-291: the authors should explain more explicitly the potential link between dietary items of each type of environment and the specific functional categories that are found associated with these environments. Are dietary items found in stream environments (amphipods, insects) more enriched in lipids than those found in ponds (ostracods, copepods, cladocerans), which would explain the differences of content in gene functional categories? Or is it better explained by abiotic factors specific to each environment?

Response: We agree that establishing such connections would be interesting, and we have spent some time screening the pertinent literature. Unfortunately, we found that there is only limited information on nutritional quality of macroinvertebrates in general, and no such data at all on the specific taxa inhabiting the Kottenforst system. From previously published data, it furthermore seems that the nutritional value of macroinvertebrates is also quite variable through time and among rather closely related taxa (e.g., larvae of different insect groups). Therefore, without having data specifically from our study system, we would be a bit wary to make overarching connections here. And lastly, because we are generating functional information with a predictive approach we prefer not speculate about dietary-function connections. We have

therefore not made changes to the manuscript in reference to this suggestion, but will keep this suggestion in our mind for future studies; in fact, combining true metagenomic and metatranscriptomic assessments with metabolomic studies of the prey items involved would be an exciting prospect and will certainly allow gaining further insights into this system.

Minor

L119: “denovo OTUs”. In the literature, the term denovo OTU is usually employed to describe an OTU that has been called with a denovo algorithm without the help of an external database. Here, the authors are referring to “OTUs with unknown taxonomic classification”. I think this sentence should be rephrased to avoid confusion.

Response: We believe there was a possible misunderstanding here. In fact, we used denovo here to refer to OTUs clustered with a denovo algorithm, which is the second step of open reference OUT picking. We have rephrased the parenthetical phrase “(i.e. through not present in the SILVA database.)” in an effort to minimize this confusion. The phrase now reads: “(i.e., OTUs clustered by a denovo algorithm in the QIIME open-reference clustering pipeline)”. (Line 118-119)

L180: Chao1 diversity differed slightly between... => Chao1 diversity slightly differed between...

Response: This change has been made.

Mathieu Groussin

Reviewer #3 (Remarks to the Author):

The authors have addressed all reviewers' comments and I have no further comments or suggestions. I trust that all edits have been made as stated in the rebuttal letter and congratulate the authors with this new version of the manuscript.

Signed, Elisabeth Bik, Stanford University.